# Enalapril mitigates senescence and aging-related phenotypes in human cells and mice via pSmad1/5/9-driven antioxidative genes

Wencong Lyu[1†], Haochen Wang[1†], Zhehao Du[1], Ran Wei[1], Jianuo He[1], Fanju Meng[1], Jinlong Bi[1], Lijun Zhang[1], Chao Zhang[2], Yiting Guan[3*‡], Wei Tao[1*‡]

[1]The MOE Key Laboratory of Cell Proliferation and Differentiation, School of Life Sciences, Peking University, Beijing, China; [2]Kunming Institute of Zoology, Chinese Academy of Sciences, Kunming, China; [3]Zhanjiang Institute of Clinical Medicine, Central People's Hospital of Zhanjiang, Guangdong Medical University, Zhanjiang, China

*For correspondence:
ytguan@pku.edu.cn (YG);
weitao@pku.edu.cn (WT)

[†]These authors contributed equally to this work

[‡]These authors jointly supervised this work

**Competing interest:** The authors declare that no competing interests exist.

## eLife Assessment

This study provides **valuable** insights into the anti-senescence effects of enalapril, identifying pSmad1/5/9 signaling and associated antioxidant pathways as key mediators of its physiological benefits in aged mice. The authors present **solid** experimental evidence across both in vitro and in vivo systems, demonstrating improved organ function and reduced senescence markers following treatment. Overall, the work supports the repurposing potential of enalapril in aging research and expands understanding of its molecular targets.

**Abstract** Aging increases the risk of a myriad of chronic diseases, which are expensive and difficult to treat owing to their various risk factors. Repurposing existing medications has accelerated the development of therapies aimed at slowing aging. In this study, using IMR90 cells and aged mice, we revealed that enalapril, a drug widely prescribed for hypertension, can improve both cellular senescence and individual health. Mechanistically, phosphorylated Smad1/5/9 act as pivotal mediators of the anti-senescence properties of enalapril. It stimulates downstream genes involved in cell cycle regulation and antioxidative defenses, facilitating cell proliferation and diminishing the production of reactive oxygen species (ROS), thus increasing the antioxidative ability of enalapril. At the organismal level, enalapril has been shown to bolster the physiological performance of various organs; it notably enhances memory capacity and renal function and relieves lipid accumulation. Our work highlights the potential of enalapril to augment antioxidative defenses and combat the effects of aging, thereby indicating its promise as a treatment strategy for aging-associated diseases and its use for healthy aging.

## Introduction

Aging is a multifaceted process influenced by numerous factors, and its intricate mechanisms are not yet fully understood. Geroscience is a burgeoning field of study in which researchers hypothesize that aging is a significant risk factor for many chronic diseases. These findings suggest that interventions targeting the mechanisms of aging might delay the onset or slow the progression of

these diseases (*Kirkland, 2016*; *López-Otín et al., 2023*). Cellular senescence, a pivotal indicator of aging, is characterized by irreversible cell cycle arrest triggered by cellular damage or stress (*Khosla et al., 2020*). During cellular senescence, cells exhibit increased activity of the lysosomal enzyme senescence-associated β-galactosidase (SA-β-gal), decreased proliferation marker Ki67 signals and increased senescence-associated secretory phenotype (SASP) secretion (*Coppé et al., 2008*; *Dimri et al., 1995*).

Senescence is the result of a complex interplay of endogenous and exogenous signaling pathways (*Cohen et al., 2022*). Currently, the activation of tumor suppressor pathways, such as p16$^{INK4a}$/RB, is deemed essential for senescence-related cessation of proliferation (*Baker et al., 2011*). The disruption of signaling pathways disrupts tissue homeostasis, leading to proinflammatory responses that are increasingly recognized as fundamental aspects of senescence (*Coppé et al., 2008*; *Shah et al., 2013*). Researchers have suggested that the expression of p16 is suppressed by inhibitor of DNA-binding (ID) proteins. The expression of the *ID1* gene is downregulated by transforming growth factor-β (TGF-β), while it is upregulated by BMP-mediated phosphorylated SMAD signaling in various cell types (*Hayashi et al., 2016*; *Liang et al., 2009*). SMADs are crucial transcriptional effector factors responsible for transmitting TGF-β/BMP signals (*Urist, 1965*). They convey signals from the TGF-β superfamily and homomeric serine/threonine kinases on the cell surface to the nucleus, directly regulating transcriptional programs critical for key events (*Miyazono et al., 2005*). Numerous studies have shown that the BMP-SMAD signaling pathway can modulate genes involved in the cell cycle, which affects the balance between cell proliferation and senescence (*Genander et al., 2014*; *Hayashi et al., 2016*). Phosphorylated SMAD plays a crucial role in the induction and maintenance of pluripotency, helping maintain tissue homeostasis during aging (*Orlowski, 2017*). However, dysregulation of BMP signaling during aging can lead to fibrosis, impaired wound healing, or excessive tissue calcification, which are detrimental to tissue function (*Lyu et al., 2018*; *Rahman et al., 2015*).

Owing to increasing insights into cellular and individual aging, novel aging mechanisms and signaling pathways have been identified, leading to the development of new therapeutic interventions. Senescent cells have recently been targeted in therapy, given their substantial impact on aging (*Childs et al., 2017*). Consequently, eliminating senescent cells through pharmacological interventions, known as 'senolytics,' may delay the onset or slow the progression of age-related diseases and enhance physiological function (*Childs et al., 2017*). Research has proven that specific senolytic drugs, such as dasatinib and quercetin, can eliminate senescent cells, reverse aging phenotypes, and increase the survival rate of mice (*Xu et al., 2018*; *Zhu et al., 2015*). However, the current senolytics are not universally effective across different cell types and display inconsistent sensitivities, thereby limiting their broad application (*Yousefzadeh et al., 2018*; *Zhu et al., 2016*). In addition to senolytics, researchers have explored pharmacological interventions that aim to ameliorate or reverse aging phenotypes without the need to eliminate senescent cells. This included both the repurposing of existing medications and the development of new ones (*Bharath et al., 2020*; *Harrison et al., 2014*). However, repurposing is particularly attractive due to its cost-effectiveness; these drugs have already undergone rigorous safety evaluations, allowing for quicker implementation in new contexts. For example, drugs such as metformin, which is typically used to treat type 2 diabetes, and rapamycin, an immunosuppressant, have shown promise in reducing aging phenotypes and extending the lifespan (*Bharath et al., 2020*; *Harrison et al., 2009*). This strategy is an innovative pharmacological intervention in which existing compounds target the aging process.

In this study, we employed a drug repurposing strategy to evaluate the anti-senescence potential of several FDA-approved drugs and identified enalapril as a promising candidate. Our investigation revealed that enalapril increases pSmad1/5/9 levels, which in turn upregulates key downstream genes involved in cell cycle regulation and antioxidative defense, thereby improving cell proliferation, reducing inflammation, and decreasing reactive oxygen species (ROS) levels. Our findings confirmed the anti-senescence mechanism of enalapril at both the cellular and organismal levels. Enalapril has the ability to ameliorate aging-related phenotypes of various organs, with notable effects on the brain, kidney, and liver. Furthermore, treating aged mice with enalapril improved spatial memory and anxiety behavior, alleviated lipid accumulation, and mitigated kidney aging-related phenotypes. Taken together, the potential of enalapril to extend the healthspan and ameliorate age-related diseases makes it a compelling candidate for research and development in the field of anti-aging therapeutics.

## Results

### Enalapril restrains cellular replicative senescence

To identify drugs that are capable of delaying cellular senescence, we established an in vitro replicative senescence model using primary human embryonic lung fibroblasts (IMR90). Compared with young cells, long-term-cultured IMR90 cells presented a characteristic senescence phenotype, characterized by increased SA-β-gal staining, increased expression of the cell cycle arrest factors p16 and p21, and reduced expression of the proliferation marker Ki67 (*Figure 1—figure supplement 1A–C*). Additionally, the release of SASP factors that significantly alter the cellular microenvironment, including the well-described factors *IL1β*, *IL6*, and *CXCL10,* has also been reported to increase the degree of senescence (*Figure 1—figure supplement 1D*).

Employing a drug repurposing strategy, we discovered that enalapril, an antihypertensive medication, delayed senescence. Notably, enalapril treatment at various concentrations reduced SA-β-gal staining, and increased Ki67 positivity in IMR90 cells (*Figure 1A, B, D and E*). Western blot analysis confirmed a decrease in the senescence markers p16 and p21 with the optimal effect of 10 μM enalapril (*Figure 1C*). To further analyze the effects of enalapril on transcriptome dynamics upon senescence entry, RNA-seq was performed across enalapril-treated cells at different concentrations. In general, we found that the expression of SASP genes decreased significantly (*Figure 1F*). The expression of cell cycle arrest factors typically increased during senescence; however, the expression of these factors decreased after exposure to enalapril, whereas the expression of cell cycle genes related to cell proliferation increased upon treatment with enalapril at the optimal concentration of 10 μM (*Figure 1F*). We, therefore, treated the cells with 10 μM enalapril via RT-qPCR and detected that many prominent SASP genes, such as *IL1β and IL6*, were significantly reduced compared with those in the control (*Figure 1G*). Moreover, enalapril treatment increased the cell growth rate and delayed the transition to the growth arrest phase (*Figure 1H*). Our comprehensive analysis underscores enalapril's potential as an anti-senescence agent, warranting further exploration for its role in combating cellular senescence.

### The function of enalapril depends on phosphorylated Smad1/5/9

Given the potential of enalapril to decelerate senescence, we screened several important targets of the senescence-related pathway to identify the underlying mechanism of senescence entry. We found that enalapril notably elevated the phosphorylation level of Smad1/5/9 without affecting Smad2/3 phosphorylation, effectively reversing the decline of pSmad1/5/9 levels in senescent cells, suggesting a possible role for the BMP-pSmad1/5/9 pathway (*Figure 2A*, *Figure 2—figure supplement 1A, B*). To validate the pathway alterations in the gene expression profile, GSEA confirmed that enalapril treatment increased BMP signaling (*Figure 2B*). Concurrently, the level of the Smad4 protein, which is required for pSmad1/5/9 to enter the nucleus, also increased, suggesting that enalapril might facilitate the nuclear entry of pSmad1/5/9 and modulate downstream genes (*Figure 2A*; *Guglielmi et al., 2021*; *Senft et al., 2019*). Notably, the expression of Smad4 was reduced in senescent cells, supporting the notion that enalapril reverses senescence-associated defects in BMP signaling (*Figure 2—figure supplement 1*). To further assess the impact on gene regulation, we employed CUT&Tag analysis of pSmad1/5/9 and found that enalapril treatment increased the pSmad1/5/9 binding signals at the transcription start site (TSS) regions (*Figure 2C*).

To explore the potential role of pSmad1/5/9 in anti-senescence mechanisms, we experimentally manipulated its levels by supplementing with BMP4 to increase pSmad1/5/9 levels, or by reducing it using the BMP receptor inhibitor LDN193189 (LDN) and shRNA-mediated knockdown of the BMP receptor BMPR1A. Our findings revealed that BMP4 increased pSmad5 levels, which correlated with decreased p16 and p21, reduced β-gal staining and increased Ki67 positivity, indicating a mitigated senescent phenotype (*Figure 2—figure supplement 2*). In contrast, when cells were treated with the BMP receptor inhibitor LDN or subjected to *BMPR1A* knockdown, the changes in pSmad1/5/9 levels and senescence markers were completely reversed, highlighting that the suppression of pSmad1/5/9 accelerates the progression of senescence (*Figure 2—figure supplement 3*).

Encouraged by the observed association between pSmad1/5/9 and senescence, we tested whether pSmad1/5/9 mediates the anti-senescence effects of enalapril by co-treating with enalapril and either LDN or shRNA targeting *BMPR1A*. RNA-seq analysis revealed that LDN upregulated SASP genes and downregulated cell cycle genes. On the basis of these findings, the subsequent addition

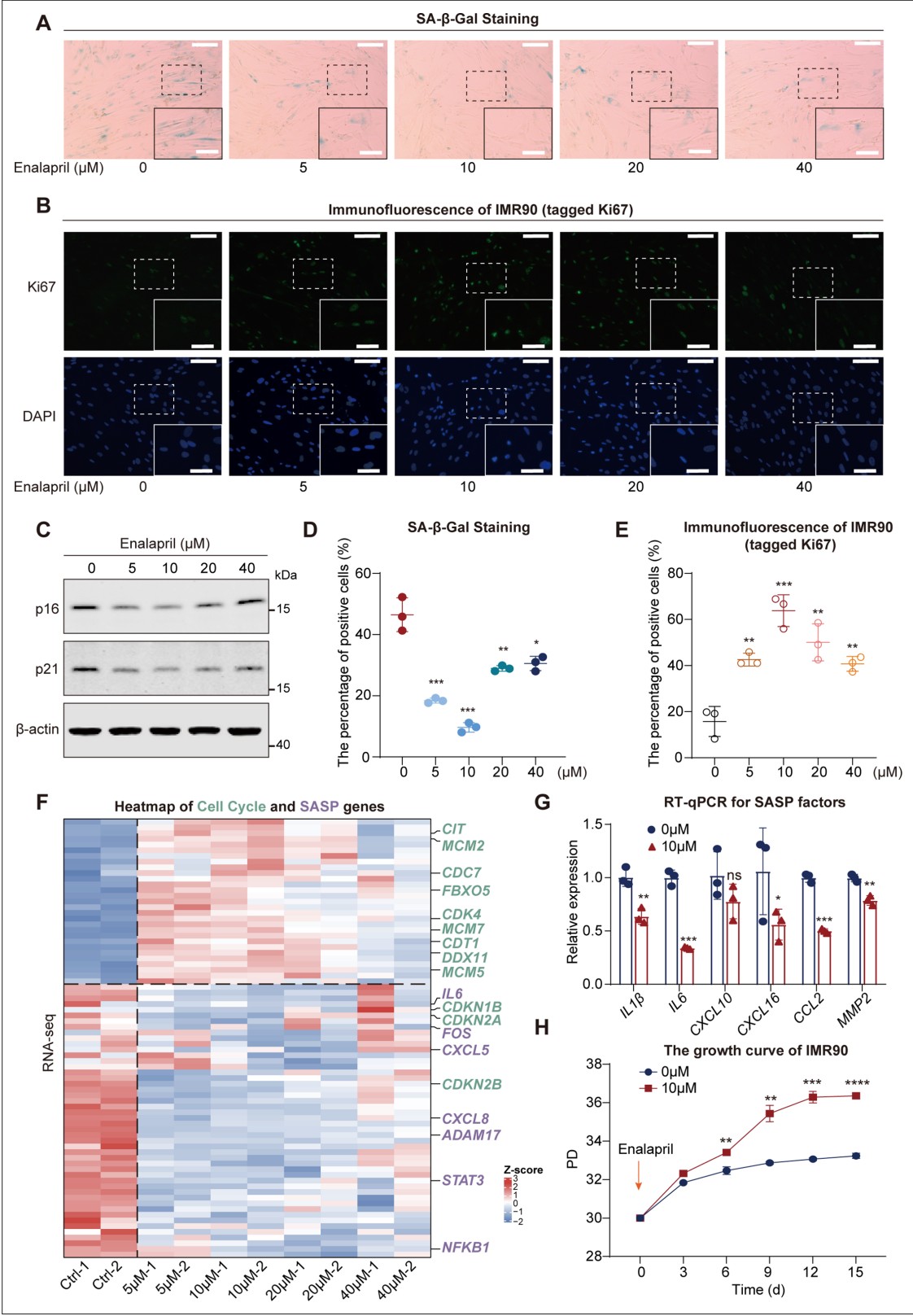

**Figure 1.** Enalapril alleviates cellular senescence. (**A, D**) Senescence-associated β-galactosidase (SA-β-Gal) staining (**A**) and statistical analysis of SA-β-Gal ratios (**D**) in IMR90 cells after enalapril treatment. Scale bars, 200 μm. Enlarged scale bars, 100 μm. (**B, E**) Ki67 immunofluorescence experiment (**B**) and statistical analysis of Ki67 intensity (**E**) after enalapril treatment. Scale bars, 80 μm. Enlarged scale bars, 40 μm. (**C**) Western blot analysis showing the protein levels of p16 and p21 in IMR90 cells after enalapril treatment. (**F**) Heatmap showing cell cycle-related gene (green) upregulation and

*Figure 1 continued on next page*

*Figure 1 continued*

senescence-associated secretory phenotype (SASP) gene (purple) downregulation after enalapril treatment. (**G**) RNA expression of SASP factors in IMR90 cells after enalapril treatment, with blue bars representing the control group and red bars representing the enalapril-treated group. (**H**) Growth curves of IMR90 cells after enalapril treatment. The x-axis represents the passage time, while the y-axis represents population doubling (PD). A two-tailed t-test was employed; ns indicates no significant difference, $*p<0.05$, $**p<0.01$, $***p<0.001$, $****p<0.0001$.

The online version of this article includes the following source data and figure supplement(s) for figure 1:

**Source data 1.** PDF file containing original western blots for *Figure 1C*, indicating the relevant bands and treatments.

**Source data 2.** Original files for western blot analysis displayed in *Figure 1C*.

**Figure supplement 1.** The expression of various senescence markers during IMR90 cell senescence.

**Figure supplement 1—source data 1.** PDF file containing original western blots for *Figure 1—figure supplement 1B*, indicating the relevant bands and treatments.

**Figure supplement 1—source data 2.** Original files for western blot analysis displayed in *Figure 1—figure supplement 1B*.

---

of enalapril did not reverse these changes, which suggests that the anti-senescence function of enalapril relies on pSmad1/5/9 (*Figure 2D*). Furthermore, western blot results also revealed that in the presence of LDN, neither enalapril nor BMP4 restored Smad1/5/9 phosphorylation or the expression of p16 and p21 (*Figure 2E*). Similarly, β-gal and Ki67 staining also revealed that enalapril failed to counteract the senescence-promoting effects induced by either LDN treatment or BMPR1A knockdown (*Figure 2F and G*, *Figure 2—figure supplement 3F, G*). Collectively, these results indicate that elevated pSmad1/5/9 could increase the expression of critical genes, thereby exerting a potent anti-senescence effect.

## pSmad1/5/9 modulates the cell cycle and SASP secretion by upregulating *ID1*

The ID family, which is recognized as a key downstream target of pSmad1/5/9, serves as a biomarker for Smad1/5/9 phosphorylation levels (*Genander et al., 2014*; *Hayashi et al., 2016*; *Ramachandran et al., 2018*; *Sun et al., 2022*; *Ying et al., 2003*). Coincidentally, we found that *ID1* was the top-ranked transcription factor that pSmad1/5/9 bound to its promoter region via CUT&Tag analysis after enalapril treatment (*Figure 3A*). Moreover, IGV analysis confirmed that enalapril significantly increased pSmad1/5/9 enrichment at the *ID1* promoter (*Figure 3B*). Simultaneously, the expression levels of both *ID1* and *ID2* were elevated (*Figure 3C and D*). Conversely, inhibiting pSmad1/5/9 with LDN or shRNA targeting BMPR1A led to decreased *ID1* and *ID2* levels, confirming the regulatory link between pSmad1/5/9 and the ID family (*Figure 3E*, *Figure 2—figure supplement 3D*). This finding indicated that the ID family was indeed a downstream gene regulated by pSmad1/5/9, and the increase in ID1 induced by enalapril further confirmed that enalapril could upregulate pSmad1/5/9 levels, thereby affecting the expression of downstream genes.

Prior studies have indicated that the ID family can suppress p16 and p21 expression, thus inhibiting cell cycle arrest and delaying cellular senescence (*Hayashi et al., 2016*; *Lyden et al., 1999*; *Ying et al., 2003*). This prompted us to examine ID protein expression during senescence, where we observed a decrease in ID1 and ID2 levels (*Figure 3—figure supplement 1A*). The knockdown of *ID1* and *ID2* via shRNA or an ID1 inhibitor resulted in a decreasing trend of ID1 or ID2 expression and a significant increase in p16 and p21 expression (*Figure 3F, I*). In addition, while β-gal increased the proportion of positively stained cells, the Ki67 proliferation signal decreased nearly fourfold after the knockdown of *ID1* or *ID2*, resembling the changes observed during senescence (*Figure 3—figure supplement 1B, C*). Given that enalapril reduced SASP factors while LDN increased them, we investigated whether ID1, which acts as a transcriptional repressor downstream of pSmad1/5/9, could modulate SASP expression. Indeed, the expression of traditional SASP factors, such as *IL1β* and *IL6*, which decrease with enalapril treatment, significantly increased after *ID* knockdown (*Figure 3G*). By overlapping the genes downregulated by enalapril with those upregulated upon *ID* knockdown, several SASP genes, as well as *CDKN2A* and *CDKN2B*, were identified (*Figure 3H*). Consistent with these findings, we found that SASP factor expression also increased in *ID* knockout mouse bone marrow cells in previously reported data (*Figure 3—figure supplement 1E*; *Fei et al., 2023*). Importantly, *ID* knockdown did not affect pSmad1/5/9 levels, confirming that ID is a downstream effector of pSmad1/5/9, influencing the expression of genes such as p16, p21, and SASP-related genes (*Figure 3J*).

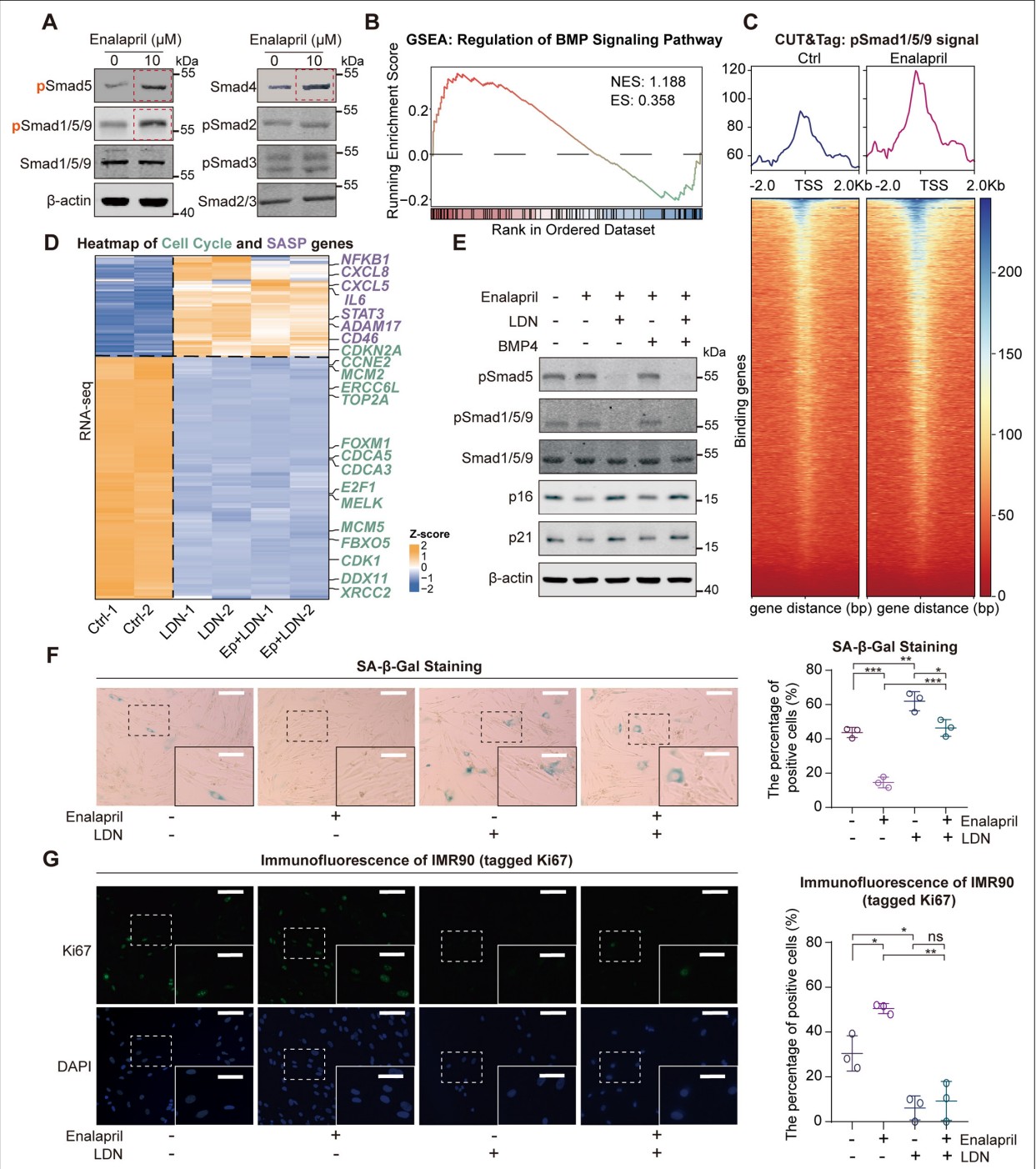

**Figure 2.** Phosphorylated Smad1/5/9 mediates the anti-senescence effect of enalapril. (**A**) Western blot results for the detection of pSmad5, pSmad1/5/9, and other proteins after enalapril treatment. (**B**) GSEA showing the enrichment of the BMP signaling pathway after enalapril treatment. NES, normalized enrichment score. ES, enrichment score. (**C**) Profile of pSmad1/5/9 enrichment at the TSS regions in the control group (Ctrl) and the enalapril-treated group (Enalapril). (**D**) Heatmap of cell cycle (green) and SASP-related genes (purple) in the control group (Ctrl), BMP receptor inhibitor-treated group (LDN193189, LDN), and enalapril and BMP receptor inhibitor-cotreated group (EP+LDN). (**E**) Western blot analysis showing the protein levels of pSmad1/5/9, p16, and p21 in response to different combinations of enalapril, a BMP receptor inhibitor (LDN193189, LDN), and BMP4. (**F**) Senescence-associated β-galactosidase (SA-β-Gal) staining (left) and corresponding ratio statistical chart (right) of IMR90 cells treated with different combinations of enalapril, a BMP receptor inhibitor (LDN193189, LDN), and BMP4. Scale bars, 200 μm. Enlarged scale bars, 100 μm. (**G**) Ki67 immunofluorescence experiment (left) and intensity statistical chart (right) of IMR90 cells treated with different combinations of enalapril, a BMP receptor inhibitor (LDN193189, LDN), and BMP4. Scale bars, 80 μm. Enlarged scale bars, 40 μm. A two-tailed t-test was employed; ns indicates no significant difference, *$p<0.05$, **$p<0.01$, ***$p<0.001$.

*Figure 2 continued on next page*

*Figure 2 continued*

The online version of this article includes the following source data and figure supplement(s) for figure 2:

**Source data 1.** PDF file containing original western blots for *Figure 2A and E*, indicating the relevant bands and treatments.

**Source data 2.** Original files for western blot analysis displayed in *Figure 2A and E*.

**Figure supplement 1.** Alterations in critical regulators of cellular senescence.

**Figure supplement 1—source data 1.** PDF file containing original western blots for *Figure 2—figure supplement 1*, indicating the relevant bands and treatments.

**Figure supplement 1—source data 2.** Original files for western blot analysis displayed in *Figure 2—figure supplement 1*.

**Figure supplement 2.** pSmad1/5/9 alleviates cellular senescence phenotypes.

**Figure supplement 2—source data 1.** PDF file containing original western blots for *Figure 2—figure supplement 2A*, indicating the relevant bands and treatments.

**Figure supplement 2—source data 2.** Original files for western blot analysis displayed in *Figure 2—figure supplement 2A*.

**Figure supplement 3.** Reduction of pSmad1/5/9 correlates with cellular senescence.

**Figure supplement 3—source data 1.** PDF file containing original western blots for *Figure 2—figure supplement 3A, D*, indicating the relevant bands and treatments.

**Figure supplement 3—source data 2.** Original files for western blot analysis displayed in *Figure 2—figure supplement 3A,D* .

After validating the prosenescence effects of *ID* through shRNA targeting *ID* or the use of small molecule inhibitors, we next examined whether the anti-senescence effects of enalapril are mediated by ID proteins. The continued addition of enalapril to these *ID*-knockdown cells did not significantly alter β-gal or Ki67 signals compared with those in the *ID*-knockdown group without enalapril, implying that the benefits of enalapril may be ID dependent (*Figure 3K–L*, *Figure 3—figure supplement 1D*). Taken together, these findings indicate that enalapril elevates pSmad1/5/9 levels, which in turn upregulate *ID1* expression. This dual action has two key implications: first, it confirms that enalapril can modulate the expression of pSmad1/5/9 downstream genes. Second, by upregulating *ID1*, enalapril can suppress the expression of p16, p21, and SASP-related genes, thereby potentially restoring cell proliferation and reducing the expression of inflammatory factors. These findings suggest that the anti-senescence mechanism of enalapril involves the pSmad1/5/9-ID1 axis, highlighting a novel pathway for its therapeutic effects on aging.

## Enalapril has antioxidative effects through pSmad1/5/9

Since an increase in pSmad1/5/9 after enalapril treatment was observed and pSmad1/5/9 is a transcription factor, to understand the in-depth mechanism of enalapril regulation, we further explored the downstream genes of pSmad1/5/9. Analysis of CUT&Tag-upregulated peaks after enalapril treatment revealed that enalapril treatment led to enrichment of pSmad1/5/9 in the GO pathway associated with DNA replication, the cell cycle, and the oxidative stress response, all of which are crucial for cell growth (*Figure 4A*). Moreover, transcriptome analysis revealed that the genes whose expression increased after enalapril treatment were also enriched in pathways related to the response to oxidative stress (*Figure 4A*). Notably, genes upregulated by enalapril in response to oxidative stress included antioxidative enzymes, electron transport chain components, and lipid oxidation inhibitors, with a significant increase in pSmad1/5/9 at their TSS regions (*Figure 4B and C*). Specifically, we also examined the promoter regions of antioxidative genes, such as the antioxidative enzyme *PRDX5* and the lipid oxidation inhibitor *TXN*, and found increased pSmad1/5/9 enrichment along with upregulated gene expression (*Figure 4D, F–H*). During cellular senescence, a decrease in these antioxidative genes exacerbates oxidative stress, reduces energy metabolism, and increases lipid oxidation, leading to the release of inflammatory cytokines (*Figure 4E*; *Coppé et al., 2008*).

It has been reported that changes in the expression of antioxidative genes can affect the ROS level of cells; thus, we then used DCFH-DA probes to detect ROS levels in cells (*Coppé et al., 2008*). The results revealed that cellular senescence increased ROS, whereas enalapril slowed the increase in ROS, indicating its ability to upregulate antioxidative genes and reduce ROS, thus affecting cellular senescence (*Figure 4I*). Importantly, the continued addition of BMP receptor inhibitors LDN reversed the beneficial effects of enalapril on ROS and decreased pSmad1/5/9 enrichment at the promoters of these antioxidative genes (*Figure 4G, I*). Similarly, shRNA-mediated knockdown of *BMPR1A* also led

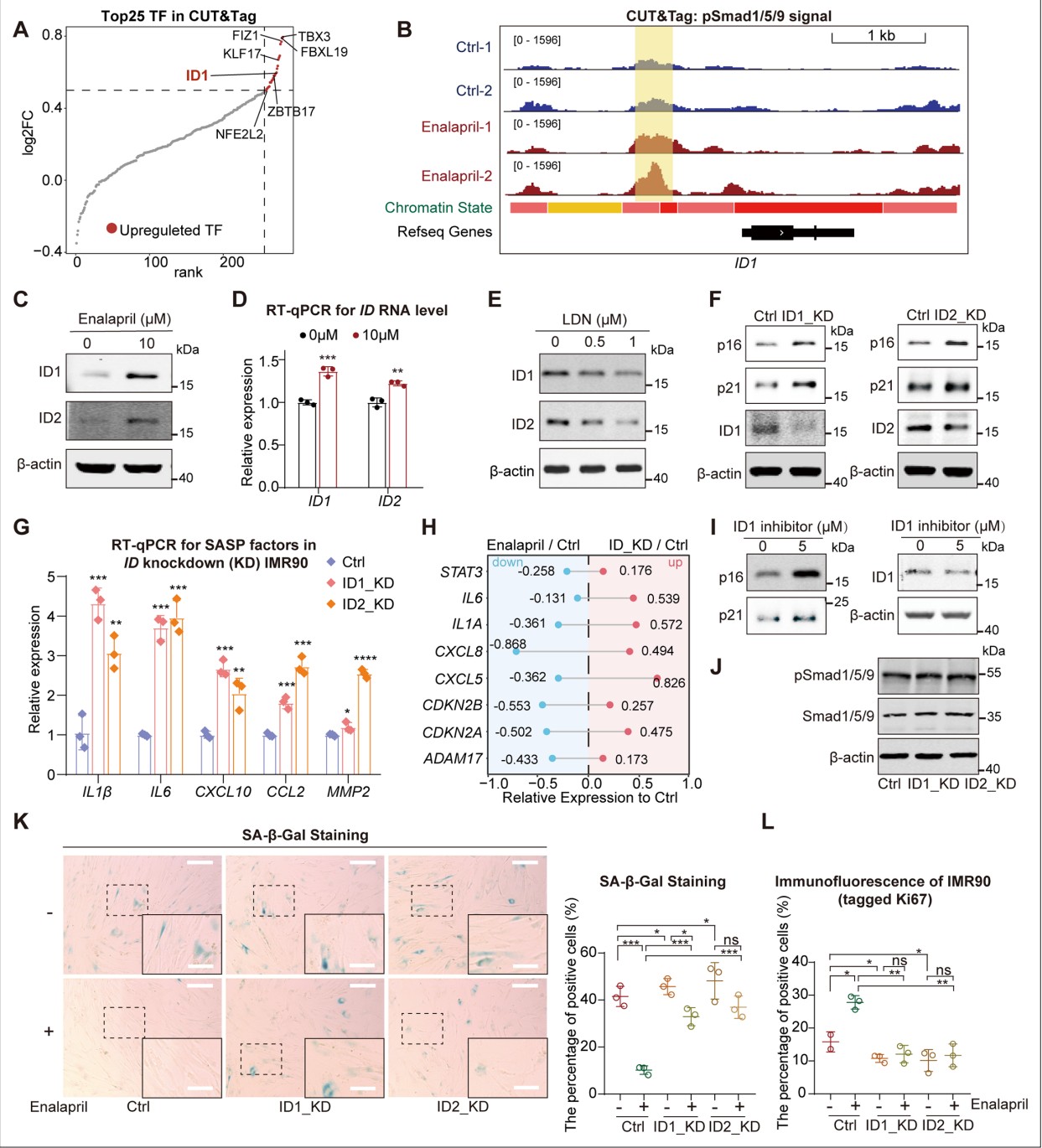

**Figure 3.** pSmad1/5/9 regulate inhibitor of DNA-binding proteins (ID)-induced inhibition of p16, p21, and senescence-associated secretory phenotype (SASP). (**A**) Changes in the peaks of all transcription factors identified by CUT&Tag following enalapril treatment, with red dots indicating transcription factors with increased peak intensity (upregulated TFs). (**B**) Integrative Genomics Viewer (IGV) showing pSmad1/5/9 signals near the *ID1* region between the control group (Ctrl) and the enalapril treatment group (Enalapril). The vertical yellow boxes indicate regions with increased signal intensity. (**C**) Western blot analysis showing the changes in the protein levels of ID1 and ID2 following enalapril treatment. (**D**) Changes in the RNA levels of ID1 and ID2 following enalapril treatment. (**E**) Western blot analysis showing the protein levels of ID1 and ID2 following treatment with a BMP receptor inhibitor (LDN193189, LDN). (**F, I**) Western blot showing the changes in the protein levels of p16 and p21 following the knockdown of *ID1* or *ID2* (**F**) or the inhibition of ID1 (**I**). (**G**) RNA expression of SASP factors after *ID* knockdown, with pink representing *ID1* knockdown and orange representing *ID2* knockdown. (**H**) Normalized average RNA expression levels of selected SASP factors and cell cycle arrest factors after enalapril treatment and *ID* knockdown. Positive values indicate upregulation, while negative values indicate downregulation. The values represent the expression levels relative to those of the Ctrl. (**J**) Western blot analysis showing pSmad1/5/9 levels following *ID1* and *ID2* knockdown. (**K**) Senescence-associated β-galactosidase (SA-β-Gal) staining (left) and SA-β-Gal ratio quantification (right) in the control group (Ctrl) and *ID*-knockdown groups (ID1_KD, ID2_KD) with or without

*Figure 3 continued*

enalapril treatment. Scale bars, 200 μm. Enlarged scale bars, 100 μm. (**L**) Ki67 immunofluorescence intensity in the control group (Ctrl) and *ID* knockdown groups (ID1_KD, ID2_KD), with or without enalapril treatment. A two-tailed t-test was employed, ns indicates no significant difference, *$p<0.05$, **$p<0.01$, ***$p<0.001$.

The online version of this article includes the following source data and figure supplement(s) for figure 3:

**Source data 1.** PDF file containing original western blots for *Figure 3C, E, F, I and J*, indicating the relevant bands and treatments.

**Source data 2.** Original files for western blot analysis displayed in *Figure 3C, E, F, I and J*.

**Figure supplement 1.** Knockdown of inhibitor of DNA-binding protein (*ID*) accelerates cellular senescence.

**Figure supplement 1—source data 1.** PDF file containing original western blots for *Figure 3—figure supplement 1A*, indicating the relevant bands and treatments.

**Figure supplement 1—source data 2.** Original files for western blot analysis displayed in *Figure 3—figure supplement 1A*.

to decreased expression of antioxidative genes such as *PRDX5* and *TXN* (*Figure 2—figure supplement 3D*). Collectively, these results reinforce the notion that enalapril promotes the upregulation of antioxidative genes through pSmad1/5/9, reduces ROS levels, and alleviates oxidative stress, thus playing a crucial role in delaying senescence.

## Enalapril upregulates pSmad1/5/9 and improves physiological function in mice

Enalapril, which is known for its anti-senescence potential, was further investigated for its impact at the organ and individual levels in 12-month-old mice (*Figure 5A*). After continuous feeding at a dosage of 30 mg/kg/day for 3 months, we conducted a comprehensive RNA-seq analysis of six key organs—brain, kidney, liver, heart, muscle, and spleen—to assess the impact of enalapril on the transcriptome. We found that enalapril induced organ-specific gene expression changes, thereby affecting the physiological functions of these genes. Moreover, GSEA revealed upregulated pathways related to forebrain and kidney development, long-chain fatty acid metabolism, and other processes, indicating that enalapril has a beneficial effect on organ functions (*Figure 5—figure supplement 1A*). Additionally, enalapril reduced the expression of SASP factors in multiple organs, particularly the CCL, CXCL, and MMP families (*Figure 5B*). Serum SASP array analysis also revealed a decrease in the secretion of certain SASP proteins, such as CCL6, CCL11, CXCL13, IGFBP-1, and MMP3, suggesting that enalapril has anti-inflammatory effects in mice (*Figure 5C*).

Given the known upregulation of pSmad1/5/9 levels and antioxidative genes in cellular models by enalapril, we measured Smad1/5/9 phosphorylation and the expression of the antioxidative genes *Txn1*, *Txn2*, and *Prdx5* in the tested organs. We observed increased pSmad1/5/9 levels, especially in the brain, kidney, and liver, along with upregulated *Id1* and antioxidative genes, notably *Txn1* and *Txn2* (*Figure 5D*, *Figure 5—figure supplement 1B*). Further analysis of BMP signaling and antioxidative stress-related pathways in these organs revealed increased gene expression after enalapril treatment, indicating the activation of this pathway (*Figure 5E*). Building on these findings, intraperitoneal injection of the BMP inhibitor LDN reversed the effects of enalapril. Specifically, SASP factor levels in serum, which were reduced by enalapril, were significantly increased. Additionally, the elevated pSmad1/5/9 and antioxidative genes in the brain, kidney, and liver were markedly downregulated upon LDN treatment (*Figure 5—figure supplement 2A, B*). These findings align with our cellular findings, implying that enalapril enhances the pSmad1/5/9-antioxidative gene axis to increase organ physiological function and delay individual aging.

## Enalapril alleviates aging-related behavior in naturally aged mice

Building on our findings of pSmad1/5/9 activation and antioxidative genes upregulation in multiple organs, we further investigated the broader phenotypic effects of enalapril in mice. After three months of enalapril feeding, behavioral experiments, serum tests, and tissue section staining were performed. Notably, enalapril had no significant effect on body weight in the mice, suggesting that the drug has no overt toxic side effects (*Figure 6A*). However, enalapril improved the decrease in age-related mobility, as evidenced by increased grip strength and extended time in the rotarod fatigue experiments, suggesting increased muscle strength and endurance (*Figure 6D and E*).

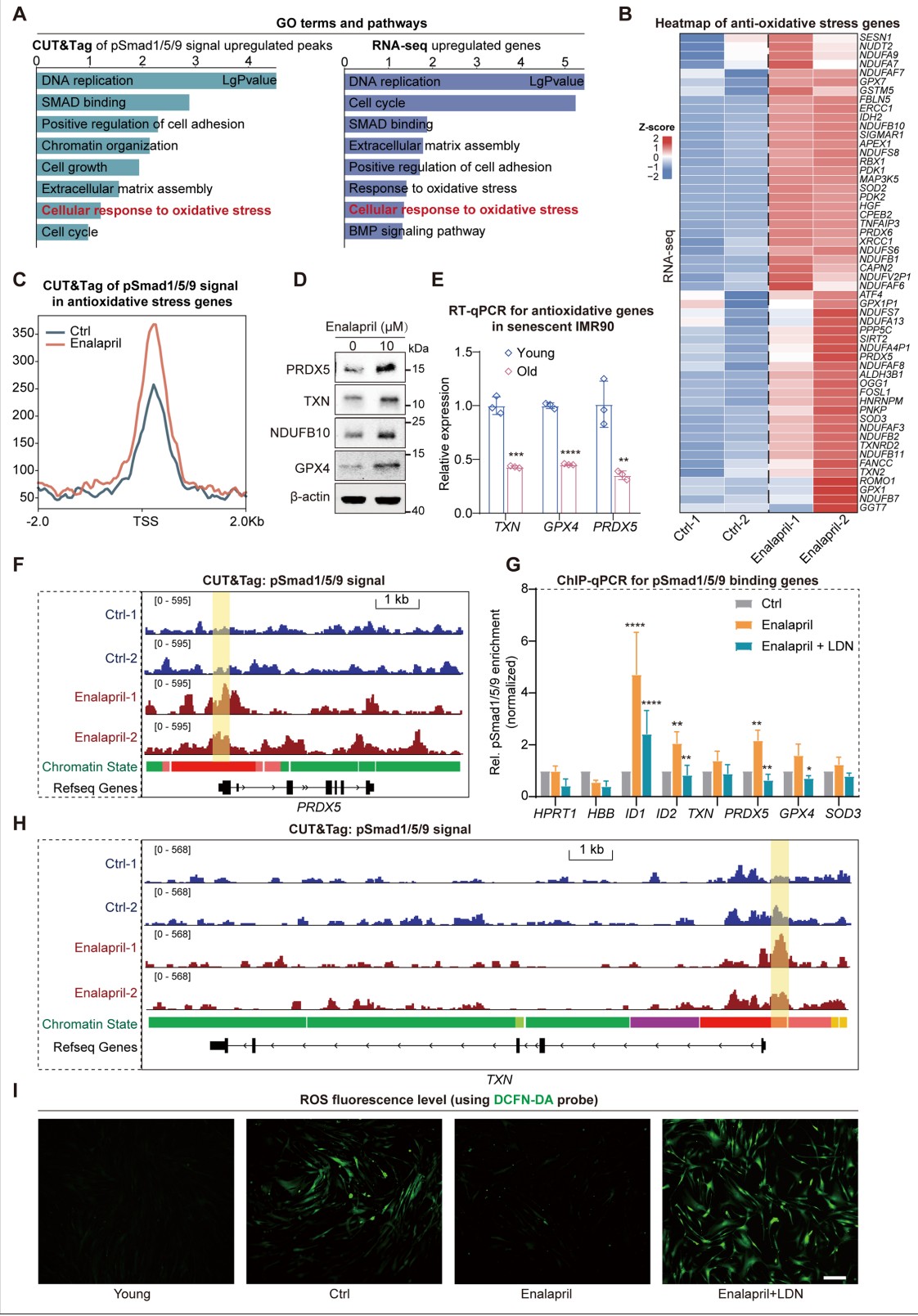

**Figure 4.** pSmad1/5/9 increase the expression of antioxidative genes. (**A**) Bar plot showing the enriched Gene Ontology (GO) terms and pathways associated with the CUT&Tag-upregulated peaks (left) and the upregulated RNA-seq genes (right) following enalapril treatment. (**B**) Heatmap showing the changes in the RNA expression of antioxidative genes. (**C**) Profile plot showing the increase in the pSmad1/5/9 binding signal of antioxidative genes in the transcription start site (TSS) region following enalapril treatment. (**D**) Western blot analysis showing the protein levels of representative

*Figure 4 continued on next page*

*Figure 4 continued*

antioxidative genes after enalapril treatment. (**E**) Relative RNA levels of *TXN*, *GPX4*, and *PRDX5* during cellular senescence. The blue bars represent young cells, and the red bars represent senescent cells. (**F, H**) Integrative Genomics Viewer (IGV) showing pSmad1/5/9 signals near the *PRDX5* (**F**) and *TXN* (**H**) regions between the control group (Ctrl) and the enalapril treatment group (Enalapril). The vertical yellow boxes indicate regions with increased signal intensity. (**G**) ChIP-qPCR results showing pSmad1/5/9 levels at many peaks following enalapril or combination treatment with enalapril and LDN193189. The y-axis represents the normalized pSmad1/5/9 signals relative to 10% input. pSmad1/5/9 enrichment for *HPRT1* and *HBB* served as a negative control, and pSmad1/5/9 enrichment for *ID1* and *ID2* served as positive controls, as previously described. A two-tailed t-test was employed, \*$p<0.05$, \*\*$p<0.01$, \*\*\*$p<0.001$, \*\*\*\*$p<0.0001$. (**I**) Detection of reactive oxygen species (ROS) fluorescence levels via the DCFH-DA probe in young cells, senescent cells (Ctrl), cells treated with enalapril, and cells treated with a combination of enalapril and LDN193189. Scale bars, 200 μm.

The online version of this article includes the following source data for figure 4:

**Source data 1.** PDF file containing original western blots for *Figure 4D*, indicating the relevant bands and treatments.

**Source data 2.** Original files for western blot analysis displayed in *Figure 4D*.

During the aging process, the brain undergoes varying degrees of functional decline. To assess the impact of enalapril on brain function, we conducted Y-maze and open field tests. The Y-maze, which is frequently employed to assess short-term memory in mice by measuring spontaneous alternation as a marker of spatial working memory, indicates notable improvements. Under normal circumstances, mice prefer to explore new maze branches rather than return to previously visited branches. Several brain regions, including the hippocampus, septum, basal forebrain, and prefrontal cortex, contribute to this task, and the age-related decline in spontaneous alternation ability in mice is associated with degeneration or lesions in brain regions such as the hippocampus (*Radulescu et al., 2021*). Enalapril promoted a substantial increase in the rate of spontaneous alternation in the mice, indicating enhanced spatial memory (*Figure 6B and C*). As aging progresses, anxiety-like behaviors often increase with age in mice (*Radulescu et al., 2021*). The open-field test, a valuable tool for assessing autonomous and exploratory behaviors in new environments, demonstrated that enalapril increased the number of central area crossings, suggesting a reduction in aging-related anxiety (*Figure 6F and G*). However, LDN treatment reversed the improvements in muscle endurance, spatial memory, and reduced anxiety behaviors conferred by enalapril (*Figure 6—figure supplement 1A–F*). In conclusion, these results suggest that enalapril has the potential to mitigate the deterioration of various behavioral aspects associated with aging through the activation of pSmad1/5/9.

## Enalapril ameliorates aging-related pathological changes in aged mice

The results of these behavioral tests provided valuable insights. To further explore the effects of enalapril on pathological changes, our investigation was extended to serum analyses. This comprehensive analysis encompassed the evaluation of vital organ function, including the liver, kidneys, and heart, as well as an assessment of lipid content, specifically triglycerides (TGs), low-density lipoprotein cholesterol (LDL), and high-density lipoprotein cholesterol (HDL). After enalapril administration, there was no significant change in the serum alanine transaminase (ALT) level, indicating that no liver toxicity from enalapril occurred (*Figure 6H*). Moreover, the level of creatinine (CREA), a key marker of kidney function, was significantly decreased in enalapril-treated mice, indicating potential functional recovery and improvement of renal function (*Figure 6I*). A heart health indicator, lactate dehydrogenase (LDH), was also reduced, indicating a positive cardiac effect of enalapril (*Figure 6J*). These findings underscore the multifaceted effects of enalapril on various physiological parameters, emphasizing its potential as a therapeutic intervention in aging-related conditions.

The transcriptomic data from the enalapril treatment group revealed an increase in long-chain fatty acid metabolism in the liver, suggesting that enalapril may play a positive role in lipid metabolism (*Figure 5—figure supplement 1A*). Accordingly, we analyzed serum lipid levels in enalapril-treated mice. Typically, aging leads to a decline in lipid metabolism in the serum, causing an accumulation of triglycerides (TGs) and low-density lipoprotein (LDL) and a decrease in high-density lipoprotein (HDL), contributing to oxidative stress and conditions such as hyperlipidemia and fatty liver (*Johnson and Stolzing, 2019*). Following enalapril treatment, the mice presented significant reductions in TG and LDL levels and a noticeable increase in HDL levels (*Figure 6K–M*). These results suggest that enalapril has the potential to increase lipid metabolism in aged mice, thereby mitigating the increase in blood lipid levels.

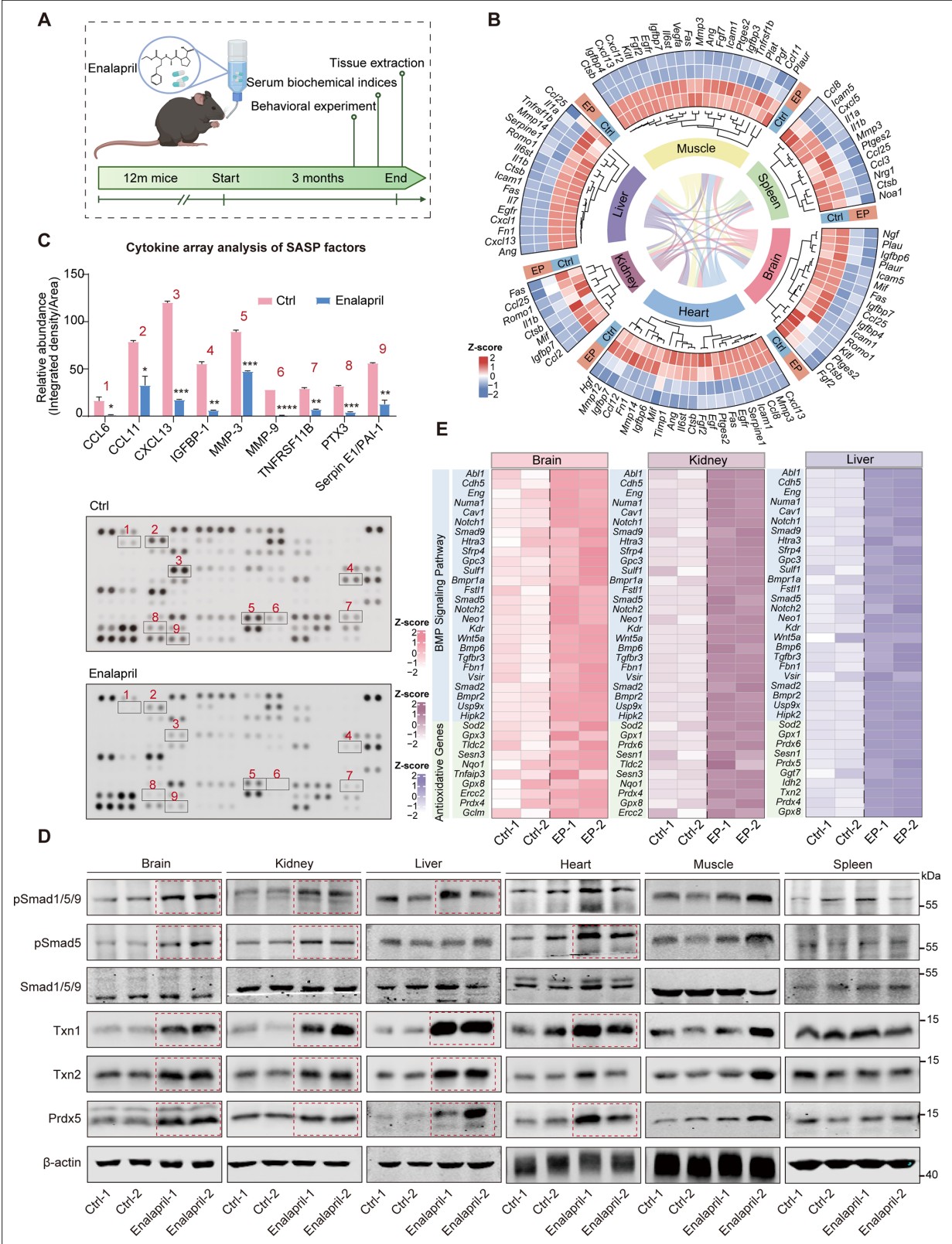

**Figure 5.** Enalapril exerts its anti-aging effects on mouse organs via pSmad1/5/9-antioxidative genes. (**A**) A schematic workflow of the mouse feeding and experimental procedures. Created with BioRender.com. (**B**) Heatmap of RNA levels of senescence-associated secretory phenotype (SASP) factors in various organs following enalapril treatment (EP). (**C**) Cytokine array analysis of secreted proteins (below) and relative quantitation (above) of SASP factors in control serum (Ctrl) and enalapril-treated serum (Enalapril). (**D**) Western blot analysis showing the protein levels of pSmad1/5/9

*Figure 5 continued on next page*

*Figure 5 continued*

and antioxidative genes in various organs following enalapril treatment. (**E**) Heatmap of the RNA levels of BMP signaling pathway-related genes and antioxidative genes in the brain, kidney, and liver following enalapril (EP) treatment.

The online version of this article includes the following source data and figure supplement(s) for figure 5:

**Source data 1.** PDF file containing original western blots for *Figure 5D*, indicating the relevant bands and treatments.

**Source data 2.** Original files for western blot analysis displayed in *Figure 5D*.

**Figure supplement 1.** Effects of enalapril on the expression changes of genes related to organ physiological functions in mice.

**Figure supplement 1—source data 1.** PDF file containing original western blots for *Figure 5—figure supplement 1B*, indicating the relevant bands and treatments.

**Figure supplement 1—source data 2.** Original files for western blot analysis displayed in *Figure 5—figure supplement 1B*.

**Figure supplement 2.** Effects of reduced pSmad1/5/9 levels on the expression changes of genes in mice.

**Figure supplement 2—source data 1.** PDF file containing original western blots for *Figure 5—figure supplement 2B*, indicating the relevant bands and treatments.

**Figure supplement 2—source data 2.** Original files for western blot analysis displayed in *Figure 5—figure supplement 2B*.

To evaluate the improvements in the aging process induced by enalapril in various organ systems, we conducted pathological examinations of mouse tissues. A decrease in β-gal staining was observed in the brain, liver, and spleen of enalapril-treated mice, suggesting an alleviation of aging-associated phenotypes (*Figure 6N*, *Figure 6—figure supplement 2A–C*). Given the effects of enalapril on brain function, we further examined pathological markers in the brain. Periodic acid-Schiff (PAS) staining, which is indicative of glycogen deposition, revealed a significant reduction in glycogen near the hippocampus (*Figure 6O*, *Figure 6—figure supplement 2D*). Congo red staining, which detects amyloid protein deposition, also revealed reduced staining in the cerebral cortex (*Figure 6P*, *Figure 6—figure supplement 2E*). In the kidney, Sirius red staining, which highlights abnormal collagen fibers or fibrosis, revealed significant red staining around blood vessels in control mice. However, this staining was noticeably reduced in the kidneys of the mice treated with enalapril (*Figure 6Q*, *Figure 6—figure supplement 2F*). These findings suggest that enalapril has a beneficial effect on fibrosis in the kidneys of aged mice, which aligns with the observed improvement in plasma creatinine levels (*Figure 6I*). Furthermore, analysis of spleen, kidney, and liver sections from enalapril-treated mice revealed a reduction in Oil Red O staining, with the kidneys and liver showing the most significant changes (*Figure 6R*, *Figure 6—figure supplement 2G–I*). This finding aligned with the consistent decrease in lipid-related molecular content in plasma, indicating that enalapril has a role in reducing age-related lipid accumulation (*Figure 6K–M*). Hematoxylin and eosin (H&E) staining of the liver revealed a decrease in cavities and increased tissue density in mice treated with enalapril, suggesting its role in enhancing liver cell proliferation and regeneration (*Figure 6S*, *Figure 6—figure supplement 2J*).

In summary, enalapril significantly mitigated multiple pathological markers associated with aging, including lipid droplets, amyloid, glycogen deposition, and fibrosis, in the brain, kidneys, liver, heart, and spleen of aged mice. These comprehensive improvements contribute to an overall amelioration of their aging status.

## Discussion

Since the early 2000s, the National Institutes of Health (NIH) has been pioneering the Interventions Testing Program (ITP), an initiative aimed at harnessing existing drugs or small molecules to decelerate the aging process (*Harrison et al., 2009*; *Nadon et al., 2008*; *Strong et al., 2016*). This approach repurposes established medications for novel uses, maximizing the potential of known compounds. It has accelerated the transition of anti-aging therapies from discovery to clinical application, ensures their safety and bolsters a comprehensive anti-aging drug screening strategy.

In our study, we discovered that enalapril, a commonly prescribed ACE inhibitor for hypertension, has significant anti-senescence effects at both the cellular and organismal levels (*Figure 7*). Enalapril demonstrated a notable ability to attenuate aging-related physiological decline across multiple organs, including the brain, kidneys, and liver. This finding suggests its potential to reduce the risk of disease across multiple organs, slow the overall aging process, and extend both the healthspan and lifespan of mice. This observation aligns with previous reports on the lifespan-extending effects

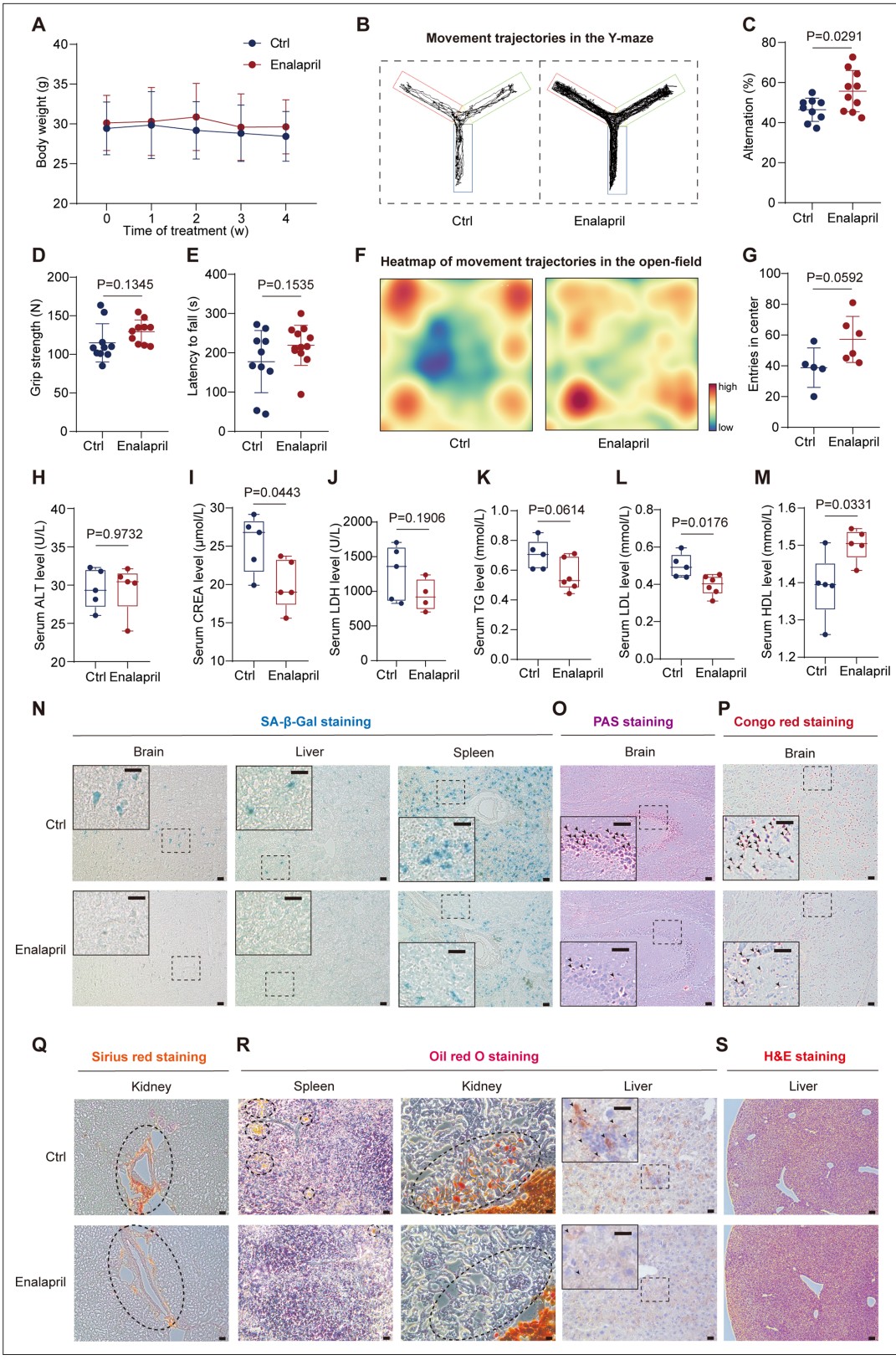

**Figure 6.** Enalapril mitigates age-related degenerative changes in aged mice. (**A**) Changes in the body weights of the mice after enalapril feeding. (**B, C**) Movement trajectory (**B**) and spontaneous alternation rate (**C**) in the Y-maze of the mice after enalapril treatment. (**D, E**) Changes in the grip strength (**D**) and rotarod fall time (**E**) of the mice after enalapril treatment. (**F, G**) Heatmap of movement trajectories (**F**) and the number of entries into the

*Figure 6 continued on next page*

*Figure 6 continued*

central area (**G**) in the open-field test after enalapril treatment. (**H-M**) Alterations in the serum levels of aspartate transaminase (AST) (**H**), creatinine (CREA) (**I**), lactate dehydrogenase (LDH) (**J**), triglycerides (TGs) (**K**), low-density lipoprotein cholesterol (LDL) (**L**), and high-density lipoprotein cholesterol (HDL) (**M**) after enalapril treatment in mice. (**N**) Senescence-associated β-galactosidase (SA-β-Gal) staining of the brain, liver, and spleen after enalapril treatment. Scale bars, 20 μm. (**O, P**) Periodic Acid-Schiff (PAS) staining (**O**) and Congo red staining (**P**) of the brain after enalapril treatment. Scale bars, 50 μm. (**Q**) Sirius red staining of the kidney after enalapril treatment. Scale bars, 50 μm. (**R**) Oil Red O staining of the spleen, kidney, and liver after enalapril treatment. Scale bars, 20 μm. (**S**) Hematoxylin and eosin (H&E) staining of the liver after enalapril treatment. Scale bars, 50 μm.

The online version of this article includes the following figure supplement(s) for figure 6:

**Figure supplement 1.** Effects of reduced pSmad1/5/9 levels on aging-related behaviors in mice.

**Figure supplement 2.** Enalapril ameliorates pathological phenotypes in aged mice.

of other ACE inhibitors, such as captopril, which have been shown to increase longevity in organisms such as *C. elegans* and mice (***Egan et al., 2024***; ***Kumar et al., 2016***; ***Strong et al., 2022***). However, our screening process revealed that enalapril outperformed captopril and other antihypertensive drugs in ameliorating senescent phenotypes, although captopril and other antihypertensive drugs also exhibited the ability to delay senescence. These findings indicate that enalapril may have unique or more potent mechanisms of action that contribute to its anti-senescence effects, warranting further mechanistic studies to better understand its molecular targets.

Mechanistically, our data revealed that enalapril enhances the phosphorylation of Smad1/5/9, a critical signaling pathway that regulates various cellular processes, including cell proliferation and the oxidative stress response. Phosphorylation of Smad1/5/9 facilitates its nuclear translocation, where it acts as a transcriptional regulator to upregulate downstream genes, such as those involved in cell cycle regulation and antioxidative responses. These findings suggest that the ability of enalapril to modulate the pSmad1/5/9 pathway contributes to its protective effects against oxidative stress, a

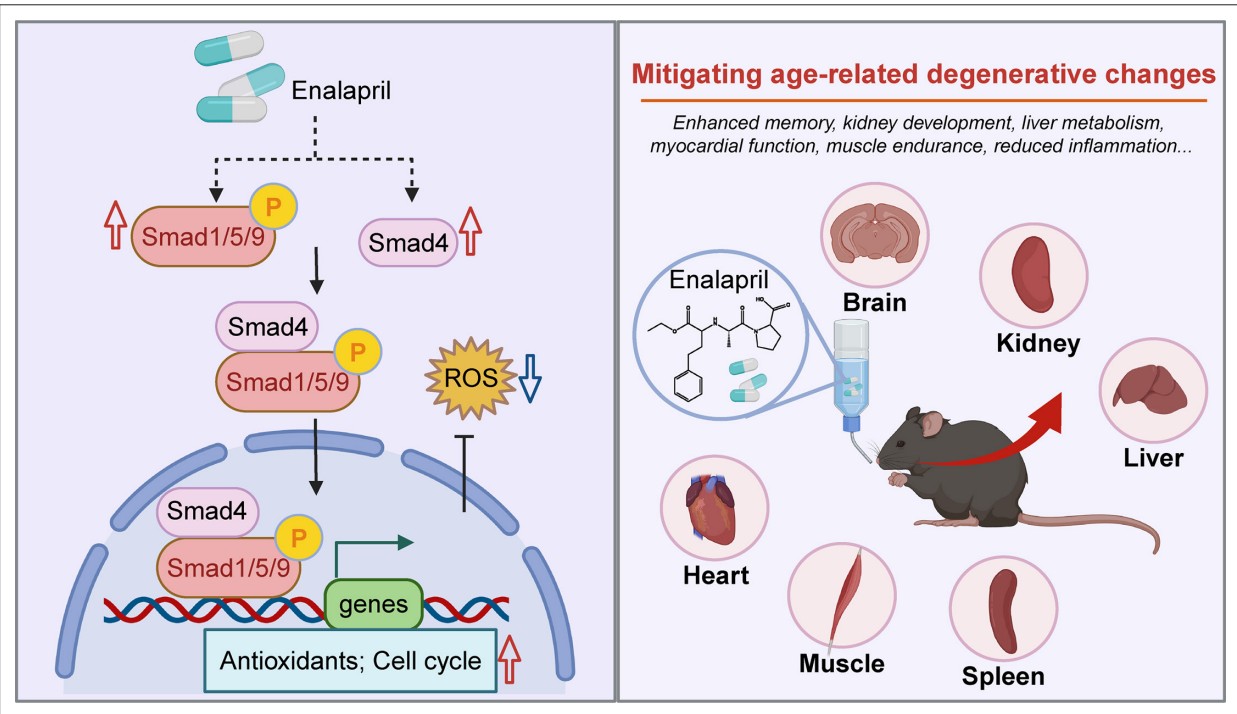

**Figure 7.** Summary schematic. It shows pSmad1/5/9-mediated combating cellular senescence of enalapril, and improvement of organ function in mice. Specifically, enalapril increases the level of pSmad1/5/9 and Smad4, promoting the expression of downstream genes, such as antioxidative and cell cycle-related genes, thereby promoting cell proliferation and reducing reactive oxygen species (ROS). Moreover, enalapril mitigates age-related degenerative changes, including enhanced memory, improved kidney function, increased liver metabolism, and other organ functions. Created with BioRender.com.

well-established hallmark of aging. By reducing ROS levels and promoting cell proliferation, enalapril effectively delays the progression of cellular senescence and reduces organismal aging-related phenotypes. This finding not only highlights the potential of phosphorylated Smad1/5/9 as a therapeutic target for anti-senescence interventions but also reveals a novel antioxidative mechanism of enalapril action.

Several FDA-approved drugs have been repurposed for anti-aging applications, demonstrating a wide range of mechanisms and effects. These include rapamycin, metformin, and NAD+ precursors. For example, rapamycin, which was originally used as an immunosuppressant to prevent organ transplant rejection, inhibits the mTOR pathway, which is associated with cell growth and metabolism (*Johnson et al., 2013*). It extends the lifespan of mice and shows potential anti-aging effects across various species (*Harrison et al., 2009*). Compared with other repurposed anti-aging drugs, enalapril offers distinct advantages because of its unique mechanism of action. As an ACE inhibitor, enalapril inherently targets the renin–angiotensin system, providing vascular protection (*Schmieder et al., 2007*; *Yamamoto and Rakugi, 2021*). Compared with repurposed drugs such as rapamycin, which may have more pronounced side effects, such as immunosuppression, limiting their long-term use, enalapril has a well-established safety profile (*Janes and Fruman, 2009*; *Saunders et al., 2001*). This can facilitate faster clinical translation for anti-aging applications. Moreover, our study revealed that enalapril reduces ROS through the pSmad1/5/9-antioxidant axis, highlighting its potential for antioxidation, a hallmark of aging that is less directly targeted by drugs such as rapamycin or metformin. Additionally, enalapril enhances multi-organ function, particularly in the brain, kidneys, and liver. Its well-established safety profile further supports enalapril as a promising candidate for clinical anti-aging applications, with the potential to complement existing therapies by promoting both vascular health and systemic antioxidative and anti-inflammatory effects.

Aging is a highly complex and multifactorial process (*Campisi, 2013*; *Jazwinski, 1996*). It involves intricate interactions among various signaling pathways and crosstalk between tissues and organs, especially during individual aging (*Jazwinski, 1996*; *Muñoz-Espín and Serrano, 2014*). Notably, enalapril is not uniform across all organs and tissues, as indicated by the differential improvement in aging phenotypes among various tissues. For example, enalapril significantly improved the physiological functions of the brain, muscle, and other organs, but the specific pathways and mechanisms by which these effects are mediated might differ among organs. This observation suggests that the tissue-specific effects of enalapril may be mediated by differential activation of signaling pathways, including those beyond the Smad1/5/9 axis, which are known to play crucial roles in aging. Future studies should focus on delineating these tissue-specific mechanisms to fully understand the scope of the anti-aging effects of enalapril. Moreover, improvements in metabolic function in the liver could have downstream effects on brain function, as metabolic dysregulation is known to exacerbate neurodegenerative diseases. Similarly, enhanced vascular health resulting from the antihypertensive properties of enalapril may contribute to improved brain perfusion, reducing the risk of age-related cognitive decline. Therefore, exploring the systemic effects of enalapril and how improvements in one organ system may benefit others should be a priority for future research. Furthermore, given the established role of enalapril as an antihypertensive drug, assessing its potential side effects when it is repurposed for anti-aging purposes is essential. Identifying strategies to minimize these side effects while optimizing their anti-aging effects is an important area of investigation. In conclusion, our study identified enalapril as a promising candidate for repurposing as an anti-senescence drug with antioxidative and promoting effects on cell proliferation. These findings open avenues for the development of targeted therapies aimed at mitigating aging and promoting healthy longevity in clinical settings, providing a foundation for guiding future research in the realm of anti-aging drug interventions.

## Materials and methods
### Cell culture
Primary human embryonic lung fibroblasts (IMR90) were obtained from ATCC (CCL-186). Cell identity was confirmed by STR analysis, and the cells were verified to be free of mycoplasma contamination. IMR90 cells were cultured in DMEM (Gibco, C11995500) supplemented with 10% fetal bovine serum (Cellmax, SA111) and 1% non-essential amino acids (Gibco, 11140050) at 37 °C and 5% $CO_2$. When

the cell density in the in vitro culture reached 80–90% (usually within 2 or 3 days), cell passage was performed.

## Drug preparation and addiction

Enalapril (MCE, HY-B0331) was dissolved in DMSO to prepare a 10 mM stock solution. When used, it was diluted to 10 µM in the culture medium. After the cells adhered to the surface, enalapril was added for treatment, and RNA and protein were collected for analysis. Recombinant BMP4 protein (MCE, HY-P7007) was dissolved in sterile ddH$_2$O to prepare a 100 µg/mL stock solution. When used, it was diluted to 50 ng/mL in the culture medium and added to the cells. The BMP receptor inhibitor (LDN193189, MCE, HY-12071) was dissolved in sterile ddH$_2$O to prepare a 1 mM stock solution. When used, it was diluted to 1 µM or 0.5 µM in the culture medium. The ID inhibitor (Pimozide, MCE, HY-12987) was dissolved in DMSO to prepare a 10 mM stock solution. When used, it was diluted to 5 µM in the culture medium. After the cells adhered to the surface, drugs were added for a 3 day treatment, and samples were collected for analysis.

## Senescence-associated β-galactosidase (SA-β-gal) staining

SA-β-gal staining was performed using a Senescence β-Galactosidase Staining Kit (Sigma–Aldrich, CAS0030). Briefly, 0.6×10$^5$ cells were seeded in a six-well plate. The cells were fixed with fixation buffer and incubated at room temperature for 10 min. After the cells were washed three times with PBS, the prepared staining mixture was added to the 6-well plate, and the plate was incubated at 37 °C overnight. The stained cells were observed and photographed under a microscope.

## Western blot

Proteins were extracted using TRIzol reagent (Invitrogen, 15596018CN) and dissolved in 1% SDS, after which SDS–PAGE was performed. The samples were transferred to nitrocellulose membranes and incubated with 5% skimmed milk at room temperature for 30 min. The primary antibodies were incubated overnight at 4 °C, followed by three washes with PBST. The membranes were then incubated with secondary antibody at room temperature for 2 hr. The band signals were visualized using an Odyssey Infrared Imaging System (Odyssey, LI-COR). All antibody information is provided in *Supplementary file 4*.

## RT-qPCR

Total RNA from cultured cells was extracted using TRIzol reagent (Invitrogen, 15596018CN) according to the manufacturer's instructions. The DEPC-treated water-dissolved RNA was subsequently reverse transcribed into cDNA using the HiScript III All-in-one RT SuperMix Perfect for qPCR (Vazyme, R333) following the product manual. The cDNA levels of specific primers were analyzed using the LightCycler 96 qPCR system (Roche), and the results were normalized to those of *ACTB*. The primers used in this analysis are listed in *Supplementary file 1*.

## RNA-seq and data processing

Total RNA was extracted using TRIzol reagent (Invitrogen, 15596018CN) following the manufacturer's instructions. The extracted total RNA was then sent to Novogene for subsequent cDNA library construction. The library was sequenced with paired-end reads on the Illumina HiSeq PE150 platform.

For data processing, FastQC (version 0.11.5) software was first used for data quality control. Raw paired-end sequencing reads were trimmed by the Trim Galore (version 0.6.10) software to remove adapter sequences and low-quality reads. The high-quality RNA-seq reads were mapped to the hg38/mm10 genome using HISAT2 (version 2.2.1) software (*Kim et al., 2019*). SAMtools (version 1.17) software was then used to convert the generated SAM files to bam files (*Li et al., 2009*). To qualify gene expression, the number of reads mapped to each gene was counted using featureCounts (version 2.0.1) software (*Liao et al., 2014*). Differentially expressed genes (DEGs) were calculated by R package DESeq2 (version 1.34.0) (*Love et al., 2014*) with the cutoff values of Benjamini-Hochberg adjusted p-value <0.05 and |log$_2$(fold change)|>0.5. The gene expression count matrix was converted into the reads per kilobase per million mapped reads (RPKM) matrix, followed by Z-score normalization for visualization.

## CUT&Tag and data processing

For the CUT&Tag experiments, $8 \times 10^5$ cells were processed using the CUT&Tag Kit (Vazyme, TD903). The cells were collected and incubated with ConA beads and then mixed with the primary antibody pSmad1/5/9 (CST, #13820) overnight at 4 °C. After incubation with the secondary antibody for 1 hr, the cells were further incubated with pA/G-Tnp for 1 hr. The transposase was then activated for DNA fragmentation. Following DNA extraction, library amplification and purification were performed. The libraries were sequenced using paired-end reads on the Illumina HiSeq PE150 platform.

For data processing, raw CUT&Tag data was trimmed using Trim Galore (version 0.6.10) software, and the clean reads were mapped to the human reference genome hg38 using Bowtie2 (version 2.5.1) software (*Langmead and Salzberg, 2012*) with the following parameters: '`--very-sensitive --no-mixed --no-discordant` -I 10 X 1000.' The SAM files were transformed into bam files via SAMtools (version 1.17) software (*Li et al., 2009*). The CUT&Tag peaks were performed using the MACS2 (version 2.2.8) with the default parameters (*Zhang et al., 2008*). Differential peaks were calculated using the R package DiffBind (version 3.4.11) with the cutoff values of adjusted p-value <0.05 and |log$_2$(fold change)|>0.5 and annotated using the R package ChIPpeakAnno (version 3.28.0) (*Zhu et al., 2010*). To visualize the CUT&Tag signal, the reads per kilobase per million mapped reads (RPKM) were calculated in each 50 bp bin size using DeepTools (version 3.5.2) software (*Ramírez et al., 2014*), and the RPKM matrix was normalized using the Z score algorithm.

## ChIP-qPCR

At room temperature, approximately $2 \times 10^6$ IMR90 cells were cross-linked with 1% paraformaldehyde for 10 min. The cross-linking reaction was stopped by adding 0.125 M glycine for 6 min. The cells were then harvested and lysed on ice in nuclear lysis buffer (50 mM Tris-Cl, 10 mM EDTA, 1% SDS, and protease inhibitors). Chromatin immunoprecipitation (ChIP) was performed using a Bioruptor (Diagenode) to shear the chromatin. Fragmented chromatin (25 µg) was immunoprecipitated with 5 µg of anti-pSmad1/5/9 antibody (CST, #13820) or anti-rabbit IgG (Abcam, ab171870). The immunoprecipitated DNA was quantified via RT-qPCR. The amount of DNA in the immunoprecipitates was calculated as a percentage relative to 10% of the input chromatin. All ChIP-qPCR-related primers are available in *Supplementary file 2*.

## Immunofluorescence

The slides were washed twice in PBS, fixed with 4% PFA for 10 min, and washed again three times with PBS. Subsequently, permeabilization was performed via incubation with 0.5% Triton X-100 for 10 min, followed by three additional washes with PBS. Next, blocking was performed with 1% BSA, and after blocking, the slides were washed three times with PBS. The primary antibody (Ki67, Abcam, ab15580) was then diluted at a ratio of 1:500 in PBST (PBS +0.1% Tween 20) and incubated overnight at 4 °C. Afterward, the slides were washed four times with PBST, with each wash lasting 5 min. The secondary antibody was diluted at a ratio of 1:500 in PBST (1× PBS + 0.1% Triton X-100) and incubated at room temperature for 1–1.5 hr. After three washes with PBST, a final wash with PBS was conducted. DAPI was diluted to a final concentration of 1–2 ng/µL and incubated at room temperature for 3–5 min. Following three washes with PBS and one wash with ddH$_2$O, the slides were sealed with Fluoromount-G (SouthernBiotech, 0100–010) and covered with a coverslip.

## Lentiviral production and viral transduction

First, the pLKO.1 empty vector or vectors encoding shRNAs targeting *BMPR1A*, *ID1*, or *ID2* (*Supplementary file 3*), along with the assistant plasmids psPAX2 and pMD2.G, were cotransfected into HEK293T cells (ATCC, CRL-3216). After 18 hr of transfection, the medium was replaced with DMEM supplemented with 30% complement-inactivated FBS, and viral supernatants were collected after 24 hr and 48 hr of transfection. When IMR90 cells reached 70% confluence, they were infected with the packaged virus with 5 µg/mL polybrene (MCE, HY-K1057). At 36–48 hr after infection, successfully transduced cells were selected via treatment with 1 µg/mL puromycin (M&C Gene Technology, MA009) for 24–36 hr.

## Detection of reactive oxygen species (ROS)

IMR90 cells were harvested from a six-well plate. After the cells were washed, the ROS-sensitive fluorescent probe 2′,7′-dichlorofluorescin diacetate (DCFH-DA, MCE, HY-D0940) was used for detection. A 10 mM stock solution of DCFH-DA was prepared by dissolving it in anhydrous DMSO. This stock solution was subsequently diluted in serum-free DMEM to obtain a 10 μM working solution of DCFH-DA. The samples were incubated with the DCFH-DA probe at 37 °C for 30 min. After incubation, the samples were washed to remove excess probe. The fluorescence intensity was measured via a fluorescence microscope. The fluorescence intensity corresponds to the level of ROS present in the sample.

## Mice feeding

All the mouse experiments were approved by the Institutional Animal Care and Use Committee of Peking University (SYXK2019-0032, IACUC No. LSC-TaoW-2). The mice were all anesthetized during all the surgeries to minimize their suffering. 12-month-old C57BL/6 mice (purchased from Wukong Biotech Co., Ltd.) were housed in a specific pathogen-free (SPF) animal facility under controlled environmental conditions with a temperature range of 20–25°C, 55 ± 10% humidity, and a 12 hr light/dark cycle. The mice had ad libitum access to food and water. A daily dosage of 30 mg/kg of enalapril was dissolved into the mice's drinking water. In the LDN treatment group, mice received intraperitoneal injections of the BMP receptor inhibitor LDN193189 (3 mg/kg, MCE, HY-12071) every other day. Four to ten mice were included in each group.

## Forelimb grip strength test

The mice were placed on a grip strength meter, and their tails were gently pulled to allow them to grasp the pull bar with their forelimbs. The grip strength meter readings were recorded at the point of maximum force exerted by each mouse. The measurements were repeated multiple times to ensure reliable and consistent results.

## Rotarod fatigue test

The mouse was placed on the rotarod, and rotation was initiated. The device was set to an accelerating mode that increased from 4 to 40 revolutions per minute over 300 s. The time at which each mouse fell off or lost balance was recorded. The test was repeated multiple times to assess the endurance and motor performance of the mice.

## Y-maze test

The Y-maze consists of three opaque arms arranged at 120° angles relative to each other. The mice were placed in the central position of the maze and allowed to explore the three arms freely for 10 min. Each consecutive entry into one of the three arms was defined as spontaneous alternation behavior. The total number of arm entries and the number of spontaneous alternations were recorded. The percentage of spontaneous alternation behavior was then calculated.

## Open field test

The mice were removed from their cages and placed in an open field, with the central area designated. The mice were allowed to move freely within the open field for 10 min, and the number of times that they crossed the central area was recorded.

## Mouse serum extraction and analysis

After the mice were anesthetized, blood (approximately 1 mL per mouse) was collected from the eyeballs. The blood was allowed to clot and settle at room temperature. The mixture was then centrifuged at 3000 rpm for 20 min, and the upper clear layer, which was the serum, was collected. The serum was subsequently subjected to biochemical assays, including ALT, CREA, LDH, TG, LDL, and HDL tests.

## Mouse tissue extraction

After the mice were euthanized, they were perfused with prechilled PBS until no blood was expelled. The heart, liver, spleen, kidney, brain, and muscle tissues were harvested. A portion of each tissue sample was used for RNA and protein extraction for sequencing and Western blot analysis, respectively,

while another portion was fixed in 4% paraformaldehyde (PFA). The tissues were then subjected to either frozen or paraffin embedding. Various staining techniques, including β-galactosidase (β-gal) staining, periodic acid-Schiff (PAS) staining, Congo red staining, Sirius red staining, Oil Red O staining, and hematoxylin, and eosin (H&E) staining were performed.

### Pathway enrichment analysis

Gene Ontology (GO) enrichment analysis was performed with Metascape (*Zhou et al., 2019*), and gene set enrichment analysis (GSEA) was performed with the R package ClusterProfiler (version 4.2.2) (*Wu et al., 2021*). Pathways with p-values <0.05 were considered significantly enriched. Representative terms were visualized with the ggplot2 (version 3.3.6) R package and ComplexHeatmap (version 2.10.0) R package (*Gu, 2022*).

### Statistical analysis

The data are expressed as means ± SEMs. The data were analyzed via a two-tailed unpaired Student's t-test. A p-value <0.05 was considered statistically significant.

## Acknowledgements

We thank Dr. Xuehui Lyu (Peking University) and Shuo Chen (Center for Excellence in Molecular Cell Science, Chinese Academy of Sciences) for suggestions regarding data organization. We thank the National Center for Protein Sciences at Peking University for assisting with the cDNA library and the Core Facilities of Life Sciences, Peking University, for assisting with the confocal microscopy. We are also grateful to the Behavioral Platform of the Animal Center at Peking University for assisting with the behavioral experiments involving the mice. This work was supported by the National Key Research and Development Program of China (2021YFA0909300).

## Additional information

### Funding

| Funder | Grant reference number | Author |
| --- | --- | --- |
| National Key Research and Development Program of China | 2021YFA0909300 | Wei Tao |

The funders had no role in study design, data collection and interpretation, or the decision to submit the work for publication.

### Author contributions

Wencong Lyu, Data curation, Formal analysis, Validation, Investigation, Visualization, Methodology, Writing – original draft, Project administration, Writing – review and editing; Haochen Wang, Data curation, Software, Formal analysis, Visualization, Writing – original draft, Writing – review and editing; Zhehao Du, Formal analysis, Validation; Ran Wei, Formal analysis, Methodology; Jianuo He, Validation; Fanju Meng, Jinlong Bi, Software; Lijun Zhang, Resources; Chao Zhang, Writing – original draft, Writing – review and editing; Yiting Guan, Supervision, Writing – original draft, Writing – review and editing; Wei Tao, Supervision, Funding acquisition, Writing – original draft, Writing – review and editing

### Author ORCIDs

Wencong Lyu ⬡ https://orcid.org/0009-0001-0456-7560
Haochen Wang ⬡ https://orcid.org/0009-0001-3169-6747
Chao Zhang ⬡ https://orcid.org/0000-0003-4167-4872
Yiting Guan ⬡ https://orcid.org/0000-0001-5158-0226
Wei Tao ⬡ https://orcid.org/0000-0002-1860-895X

## Ethics

All the mouse experiments were approved by the Institutional Animal Care and Use Committee of Peking University (SYXK2019-0032, IACUC No. LSC-TaoW-2). The mice were all anesthetized during all the surgeries to minimize their suffering.

Reviewer #1 (Public review): https://doi.org/10.7554/eLife.104774.3.sa1

Reviewer #2 (Public review): https://doi.org/10.7554/eLife.104774.3.sa2

Author response https://doi.org/10.7554/eLife.104774.3.sa3

## Additional files

### Supplementary files

Supplementary file 1. List of primers used in RT-qPCR.

Supplementary file 2. List of primers used in pSmad1/5/9 ChIP-qPCR.

Supplementary file 3. List of shRNA target sequences used in knockdown experiment.

Supplementary file 4. List of antibodies.

MDAR checklist

### Data availability

All related sequencing data have been uploaded to the NCBI Gene Expression Omnibus and are accessible through accession numbers GSE277861 and GSE277862.

The following datasets were generated:

| Author(s) | Year | Dataset title | Dataset URL | Database and Identifier |
|---|---|---|---|---|
| Lyu W, Wang H | 2025 | Enalapril mitigates senescence and aging-related phenotypes by targeting antioxidative genes via phosphorylated Smad1/5/9 [RNA-seq] | https://www.ncbi.nlm.nih.gov/geo/query/acc.cgi?acc=GSE277861 | NCBI Gene Expression Omnibus, GSE277861 |
| Lyu W, Wang H | 2025 | Enalapril mitigates senescence and aging-related phenotypes by targeting antioxidative genes via phosphorylated Smad1/5/9 [CUT&Tag] | https://www.ncbi.nlm.nih.gov/geo/query/acc.cgi?acc=GSE277862 | NCBI Gene Expression Omnibus, GSE277862 |

The following previously published dataset was used:

| Author(s) | Year | Dataset title | Dataset URL | Database and Identifier |
|---|---|---|---|---|
| Fei M, Wang Y, Chang B, Wang L | 2023 | The cell-nonautonomous function of ID1 supports AML progression from the microenvironment via ANGPLT7 | https://www.ncbi.nlm.nih.gov/geo/query/acc.cgi?acc=GSE219049 | NCBI Gene Expression Omnibus, GSE219049 |

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
