## [Editor Report · eLife Assessment]

This study provides **valuable** insights into the anti-senescence effects of enalapril, identifying pSmad1/5/9 signaling and associated antioxidant pathways as key mediators of its physiological benefits in aged mice. The authors present **solid** experimental evidence across both in vitro and in vivo systems, demonstrating improved organ function and reduced senescence markers following treatment. Overall, the work supports the repurposing potential of enalapril in aging research and expands understanding of its molecular targets.

---

## [Referee Report · Reviewer #1 (Public review)]

Summary:

In this study, the authors showed that enalapril was able to reduce cellular senescence and improve health status in aged mice. The authors further showed that phosphorylated Smad1/5/9 was significantly elevated and blocking this pathway attenuated the protection of cells from senescence. When middle-aged mice were treated with enalapril, the physiological performance in several tissues, including memory capacity, renal function and muscle strength, exhibited significant improvement.

Strengths:

The strength of the study lies in the identification of pSMAD1/5/9 pathway as the underlying mechanism mediating the anti-senescence effects of enalapril with comprehensive evaluation both in vitro and in vivo.

Weaknesses:

The major weakness of the study is the in vivo data. Despite the evidence shown in the in vitro study, there is no data to show that blocking the pSmad1/5/9 pathway is able to attenuate the anti-aging effects of enalapril in the mice. In addition, the aging phenotypes mitigation by enalapril is not evidenced by the extension of lifespan. If it is necessary to show that NAC is able to attenuate enalapril effects in the aging mice. In addition, it would be beneficial to test if enalapril is able to achieve similar rescue in a premature aging mouse model.

Comments on revisions:

The revised manuscript provided additional in vivo data that addressed my questions accordingly. I think the authors have done an excellent job in demonstrating that enalapril improved physiological phenotypes in aged mice through pSmad1/5/9 pathway.

Their response to my question regarding the test in HGPS mice was not satisfactory. Premature aging and physiological aging share substantial similarities in their pathways. Given that this is not the focus of current study and the manuscript does not provide data on HGPS mice, I think this does not affect the conclusion of the current study.

---

## [Referee Report · Reviewer #2 (Public review)]

This manuscript presents an interesting study of enalapril for its potential impact on senescence through the activation of Smad1/5/9 signaling with a focus on antioxidative gene expression. Repurposing enalapril in this context provides a fresh perspective on its effects beyond blood pressure regulation. The authors make a strong case for the importance of Smad1/5/9 in this process, and the inclusion of both in vitro and in vivo models adds value to the findings. Below, I have a few comments and suggestions which may help improve the manuscript.

A major finding in the study is that phosphorylated Smad1/5/9 mediates the effects of enalapril. However, the manuscript focused on the Smad pathway relatively abruptly, and the rationale behind targeting this specific pathway is not fully explained. What makes Smad1/5/9 particularly relevant to the context of this study?

Furthermore, their finding that activation of Smad1/5/9 leads to a reduction of senescence appears somewhat contradictory to the established literature on Smad1/5/9 in senescence. For instance, studies have shown that BMP4-induced senescence involves activation of Smad1/5/8 (Smad1/5/9), leading to the upregulation of senescence markers like p16 and p21 (JBC, 2009, 284, 12153). Similarly, phosphorylated Smad1/5/8 has been shown to promote and maintain senescence in Ras-activated cells (PLOS Genetics, 2011, 7, e1002359). Could the authors provide more detailed mechanistic insights into why enalapril seems to reverse the typical pro-senescent role of Smad1/5/9 in their study?

While the authors showed that enalapril increases pSmad1/5/9 phosphorylation, what are the expression levels of other key and related factors like Smad4, pSmad2, pSmad3, BMP2, and BMP4 in both senescent and non-senescent cells? These data will help clarify the broader signaling effects.

They used BMP receptor inhibitor LDN193189 to pharmacologically inhibit BMP signaling, but it would be more convincing to also include genetic validation (e.g., knockdown or knockout of BMP2 or BMP4). This will help confirm that the observed effects are truly due to BMP-Smad signaling and not off-target effects of the pharmacological inhibitor LDN.

I don't see the results on the changes in senescence markers p16 and p21 in the mouse models treated with enalapril. Similarly, the effects of enalapril treatment on some key SASP factors, such as TNF-α, MCP-1, IL-1β, and IL-1α, are missing, particularly in serum and tissues. These are important data to evaluate the effect of enalapril on senescence.

Given that enalapril is primarily known as an antihypertensive, it would be helpful to include data on how it affects blood pressure in the aged mouse models, such as systolic and diastolic blood pressure. This will clarify whether the observed effects are independent of or influenced by changes in blood pressure.

---

## [Author Response]

The following is the authors’ response to the original reviews.

**Reviewer #1 (Public review):**
Summary:In this study, the authors showed that enalapril was able to reduce cellular senescence and improve health status in aged mice. The authors further showed that phosphorylated Smad1/5/9 was significantly elevated and blocking this pathway attenuated the protection of cells from senescence. When middle-aged mice were treated with enalapril, the physiological performance in several tissues, including memory capacity, renal function, and muscle strength, exhibited significant improvement.Strengths:The strength of the study lies in the identification of the pSMAD1/5/9 pathway as the underlying mechanism mediating the anti-senescence effects of enalapril with comprehensive evaluation both in vitro and in vivo.

Thank you for your patient reading and great efforts to advance our research! Your comments are shown in bold font below, and specific concerns have been numbered. Our point-by-point answers are provided in standard blue font, with all modifications and additions to the MS highlighted in red text.

Weaknesses:(1) The major weakness of the study is the in vivo data. Despite the evidence shown in the in vitro study, there is no data to show that blocking the pSmad1/5/9 pathway is able to attenuate the anti-aging effects of enalapril in the mice. In addition, the aging phenotypes mitigation by enalapril is not evidenced by the extension of lifespan.

Many thanks for your careful reading and valuable comments! We fully agree with this comment. In accordance with your suggestion, we administered LDN193189 to investigate its suppressive effects on pSmad1/5/9 signaling in vivo. Notably, pharmacological inhibition of pSmad1/5/9 resulted in upregulation of enalapril-suppressed SASP factors, while conversely leading to marked decrease of downstream antioxidant genes expression across multiple organ systems (Revised Fig. S7). These analyses and corresponding sentences have been added in the Result section of the revised MS (Revised Fig.S7, Lines 222–223, 444–448).

Additionally, aging-related behavioral phenotypes were also examined following pSmad1/5/9 inhibition, including decreased muscle strength and endurance, impaired spatial memory and increased anxiety behaviors (Revised Fig. S8). These analyses and corresponding sentences have been added in the Result section of the revised MS (Revised Fig.S8, Lines 476–480). Collectively, these findings demonstrate that the anti-aging effects of enalapril in mice are mediated through the pSmad1/5/9 pathway.

In this study, we focused exclusively on assessing the improvement in the health status of aged mice, which indicates that enalapril can extend the healthspan of aged mice. While we agree that lifespan extension is an important indicator of anti-aging potential, recent studies have emphasized that healthspan, rather than lifespan alone, provides a more relevant and translational measure of aging interventions, particularly in the context of chronic disease and quality of life in aged individuals (Kennedy et al., 2014; Lopez-Otin et al., 2023). Moreover, given the strong influence of genetic background, environmental factors and stochastic events on lifespan, focusing on functional rejuvenation and delayed onset of aging-related pathologies may offer a more practical and mechanistically informative approach. Our study aims to elucidate how enalapril enhances healthy phenotypes in aged mice, however, we acknowledge the critical need for direct lifespan evaluation and intend to address this limitation in subsequent research. We sincerely hope that these explanations address your concerns.

(2) If it is necessary to show that NAC is able to attenuate enalapril effects in the aging mice. In addition, it would be beneficial to test if enalapril is able to achieve similar rescue in a premature aging mouse model.

Thanks for your suggestion. We apologize for any confusion that may have arisen due to the wording in the original manuscript. N-acetylcysteine (NAC) is widely reported as an antioxidant that scavenges reactive oxygen species (ROS) (Huang et al., 2020; Zafarullah et al., 2003). In our study, enalapril was also observed to reduce ROS levels. Therefore, NAC is unlikely to antagonize the effects of enalapril in this context, as both compounds act in a similar direction with respect to oxidative stress mitigation. To avoid potential misunderstanding, we have carefully reviewed the relevant statements in the MS and revised the text to clarify this point.

We sincerely appreciate this valuable suggestion to evaluate enalapril in a premature aging mouse model; however, the premature aging mouse models represent a pathological form of aging, whereas the naturally aged mouse models used in our study reflect physiological aging processes. While we observed beneficial effects of enalapril in naturally aged mice, these effects may not necessarily extend to premature aging models due to fundamental differences in the underlying mechanisms and progression of aging. Natural aging is characterized by the gradual accumulation of cellular damage, driven by multifactorial processes such as inflammaging and mitochondrial dysfunction. In this context, enalapril appears effective, in part by modulating SASP factors and reducing oxidative stress through the BMP-Smad signaling axis (Revised Fig. 4, 5) (Lopez-Otin et al., 2023). In contrast, premature aging models are driven by distinct mechanisms like nuclear lamina defects, which may not respond similarly to BMP-Smad axis. Moreover, genetic background, strain variability, and specific model characteristics can significantly influence treatment outcomes (Mitchell et al., 2016). For instance, rapamycin extends lifespan in wild-type mice but shows limited effects on aging, underscoring the challenge of extrapolating findings across distinct aging models (Neff et al., 2013). We sincerely hope that these explanations address your concerns. Thank you again for your great efforts in advancing our research!

**Reviewer #2 (Public review):**
This manuscript presents an interesting study of enalapril for its potential impact on senescence through the activation of Smad1/5/9 signaling with a focus on antioxidative gene expression. Repurposing enalapril in this context provides a fresh perspective on its effects beyond blood pressure regulation. The authors make a strong case for the importance of Smad1/5/9 in this process, and the inclusion of both in vitro and in vivo models adds value to the findings. Below, I have a few comments and suggestions which may help improve the manuscript.

We appreciate your great efforts in advancing our research! Your comments are shown in bold font below, and specific concerns have been numbered. Our point-by-point answers are provided in standard blue font, with all modifications and additions to the MS highlighted in red text.

(1) A major finding in the study is that phosphorylated Smad1/5/9 mediates the effects of enalapril. However, the manuscript focused on the Smad pathway relatively abruptly, and the rationale behind targeting this specific pathway is not fully explained. What makes Smad1/5/9 particularly relevant to the context of this study?

Thank you for your informative guidance, and we regret for the unclear description. As stated in the MS, after we found that enalapril could improve the cellular senescence phenotype, we screened and examined key targets in important aging-related signaling pathways, such as AKT, mTOR, ERK, Smad2/3 and Smad1/5/9 (Revised Fig. S2A, Revised Fig. 2A). We found that only the phosphorylation levels of Smad1/5/9 significantly increased after enalapril treatment. Therefore, the subsequent focus of this study is on pSmad1/5/9. We sincerely hope that these explanations address your concerns.

(2) Furthermore, their finding that activation of Smad1/5/9 leads to a reduction of senescence appears somewhat contradictory to the established literature on Smad1/5/9 in senescence. For instance, studies have shown that BMP4-induced senescence involves the activation of Smad1/5/8 (Smad1/5/9), leading to the upregulation of senescence markers like p16 and p21 (JBC, 2009, 284, 12153). Similarly, phosphorylated Smad1/5/8 has been shown to promote and maintain senescence in Ras-activated cells (PLOS Genetics, 2011, 7, e1002359). Could the authors provide more detailed mechanistic insights into why enalapril seems to reverse the typical pro-senescent role of Smad1/5/9 in their study?

Many thanks for your helpful comments! The downstream regulatory network of BMP-pSmad1/5/9 is highly complex. The BMP-SMAD-ID axis has been mentioned in many studies, and its downstream signaling inhibits the expression of p16 and p21 (Hayashi et al., 2016; Ying et al., 2003). Additionally, studies have also found that the Smad1-Stat1-P21 axis inhibits osteoblast senescence (Xu et al., 2022). In our study, enalapril was found to increase the expression of ID1, which is a classic downstream target of pSmad1/5/9 (Genander et al., 2014). Therefore, pSmad1/5/9 inhibits cellular senescence markers such as p16, p21 and SASP through ID1, thereby promoting cell proliferation (Revised Fig. 3). Furthermore, we also found that pSmad1/5/9 increases the expression of antioxidant genes and reduces ROS levels, exerting antioxidant effects (Revised Fig. 4). Together, ID1 and antioxidant genes enable pSmad1/5/9 to exert its anti-senescence effects. We sincerely hope that these explanations address your concerns.

(3) While the authors showed that enalapril increases pSmad1/5/9 phosphorylation, what are the expression levels of other key and related factors like Smad4, pSmad2, pSmad3, BMP2, and BMP4 in both senescent and non-senescent cells? These data will help clarify the broader signaling effects.

Thanks for your insightful suggestions. We observed an increase in pSmad1/5/9 and Smad4 expression, while the levels of pSmad2 and pSmad3 remained unchanged after enalapril treatment (Revised Fig. 2A). Consistently, we found that the levels of pSmad1/5/9 and Smad4 were markedly reduced in senescent cells, aligning with the upregulation of these proteins by enalapril (Revised Fig. S2B). In contrast, pSmad2 and pSmad3 showed a slight increase during senescence, while BMP2 and BMP4 were slightly decreased, though these changes were not statistically significant (Revised Fig. S2B). These findings suggest that enalapril primarily exerts its effects by enhancing pSmad1/5/9 and Smad4 levels, thereby regulating downstream target genes and contributing to the restoration of a more youthful cellular state. These analyses and corresponding sentences have been added in the Result section of the revised MS (Revised Fig.S2B, Lines 303–306, 311–313).

(4) They used BMP receptor inhibitor LDN193189 to pharmacologically inhibit BMP signaling, but it would be more convincing to also include genetic validation (e.g., knockdown or knockout of BMP2 or BMP4). This will help confirm that the observed effects are truly due to BMP-Smad signaling and not off-target effects of the pharmacological inhibitor LDN.

Many thanks for your careful reading and valuable comments! We used shRNA to knockdown the BMP receptor BMPR1A, which led to a reduction in Smad1/5/9 phosphorylation (Revised Fig. S4D, E). This was accompanied by senescence-associated phenotypes, including increased expression of p16 and SA-β-gal and decreased Ki67 staining (Revised Fig. S4F, G). Notably, the addition of enalapril failed to reverse these senescence phenotypes under BMPR1A knockdown conditions, mirroring the results observed with the BMP receptor inhibitor LDN193189 (Revised Fig. S4F, G, Revised Fig. 2F, G). Furthermore, knockdown of BMPR1A also resulted in a marked decrease in the expression of downstream targets, such as ID1 and antioxidative genes (Revised Fig. S4D). These findings strongly support the notion that enalapril exerts its anti-senescence effects through BMP-Smad signaling. These analyses and corresponding sentences have been added in the Result section of the revised MS (Revised Fig.S4D–G, Lines 323–329, 335–337, 348–351, 416–418).

(5) I don't see the results on the changes in senescence markers p16 and p21 in the mouse models treated with enalapril. Similarly, the effects of enalapril treatment on some key SASP factors, such as TNF-α, MCP-1, IL-1β, and IL-1α, are missing, particularly in serum and tissues. These are important data to evaluate the effect of enalapril on senescence.

Thanks for your comments. As for the markers p16 and p21, we observed no change in p16, while the changes in p21 varied across different organs and tissues. Nevertheless, behavioral experiments and physiological and biochemical indicators at the individual level consistently demonstrated the significant anti-aging effects of enalapril (Revised Fig. 6).

We also examined the changes in SASP factors in the serum of mice after enalapril treatment. Notably, SASP factors such as CCL (MCP), CXCL and TNFRS11B showed significant decreases (Revised Fig. 5C). The expression changes of SASP factors varied across different organs. In the liver, kidneys and spleen, the expression of IL1a and IL1b decreased, while TNFRS11B expression decreased in both the liver and muscles (Revised Fig. 5B). Additionally, CCL (MCP) levels decreased in all organs (Revised Fig. 5B). We sincerely hope that these explanations address your concerns.

(6) Given that enalapril is primarily known as an antihypertensive, it would be helpful to include data on how it affects blood pressure in the aged mouse models, such as systolic and diastolic blood pressure. This will clarify whether the observed effects are independent of or influenced by changes in blood pressure.

Thanks for your comments. While enalapril is primarily recognized for its antihypertensive properties, in our experimental setting involving aged, normotensive mice, we did not observe notable changes in systolic or diastolic blood pressure following enalapril administration. This observation aligns with previous reports indicating that enalapril does not significantly affect blood pressure in similar non-hypertensive aging models (Keller et al., 2019). Based on these findings, we cautiously interpret that the beneficial effects of enalapril observed in our study are unlikely to be driven by changes in blood pressure. We sincerely hope that these explanations address your concerns. Again, thank you for the constructive comments to advance the understanding of our work!

**Reviewer #1 (Recommendations for the authors):**
This is an interesting study that reveals enalapril is able to elevate the pSmad1/5/9 pathway to reduce ROS and inflammation to improve the health status in vitro and in vivo. While the pathway is clearly shown in cells to be involved in the enalarpril-mediated mitigation of aging, little was done to demonstrate this pathway is responsible for the in vivo effects in the physiological improvements. This can be done by ROS-reduction chemicals such as NAC and also the use of BMP receptor inhibitor LDN193189 (LDN). It is critical to show the lifespan extension in enalapril-treated animals given that the significantly improved physiological functions.

Thanks very much for your constructive recommendations. This part has already been addressed in our response to the public review.

**Reviewer #2 (Recommendations for the authors):**
The term "anti-aging" appears frequently throughout the manuscript, including in the title. However, the study doesn't directly address lifespan or a comprehensive range of aging symptoms, which are also difficult to define and measure. Many of the observed effects appeared to be driven by senescence. To be more accurate, I recommend avoiding terms like "anti-aging" and "mitigates aging", and instead replacing them with more specific phrases such as "anti-senescence", "senescence reduction/suppression", or "mitigates age-related symptoms" to better reflect the scope of the study and avoid overstating the findings.

Thanks very much for your constructive recommendations. In accordance with your suggestion, we have revised all uses of the term “aging” in the MS. To facilitate review, all changes have been clearly marked in red text.

Please provide detailed information on the antibodies used, particularly those targeting pSmad1/5/9 and other Smads.

Thanks for your helpful comment. In response, we have now provided detailed information regarding the antibodies used in this study in Revised Table S4 (Revised MS, Page 120–121).